# Infant cries convey both stable and dynamic information about age and identity

Marguerite Lockhart-Bouron [1,8], Andrey Anikin [2,3,8], Katarzyna Pisanski [2,4,8], Siloé Corvin [2,5], Clément Cornec [2], Léo Papet[2], Florence Levréro[2,6], Camille Fauchon[5], Hugues Patural[1,9], David Reby[2,6,9] & Nicolas Mathevon [2,6,7,9✉]

What information is encoded in the cries of human babies? While it is widely recognized that cries can encode distress levels, whether cries reliably encode the cause of crying remains disputed. Here, we collected 39201 cries from 24 babies recorded in their homes longitudinally, from 15 days to 3.5 months of age, a database we share publicly for reuse. Based on the parental action that stopped the crying, which matched the parental evaluation of cry cause in 75% of cases, each cry was classified as caused by discomfort, hunger, or isolation. Our analyses show that baby cries provide reliable information about age and identity. Baby voices become more tonal and less shrill with age, while individual acoustic signatures drift throughout the first months of life. In contrast, neither machine learning algorithms nor trained adult listeners can reliably recognize the causes of crying.

[1] Neonatal and Pediatric Intensive Care Unit, SAINBIOSE laboratory, Inserm, University Hospital of Saint-Etienne, University of Saint-Etienne, Saint-Etienne, France. [2] ENES Bioacoustics Research Laboratory, CRNL, CNRS, Inserm, University of Saint-Etienne, Saint-Etienne, France. [3] Division of Cognitive Science, Lund University, Lund, Sweden. [4] Laboratoire Dynamique du Langage DDL, CNRS, University of Lyon 2, Lyon, France. [5] Central Integration of Pain—Neuropain Laboratory, CRNL, CNRS, Inserm, UCB Lyon 1, University of Saint-Etienne, Saint-Etienne, France. [6] Institut universitaire de France, Paris, France. [7] Ecole Pratique des Hautes Etudes, CHArt Lab, PSL Research University, Paris, France. [8] These authors contributed equally: Marguerite Lockhart-Bouron, Andrey Anikin, Katarzyna Pisanski. [9] These authors jointly supervised this work: Hugues Patural, David Reby, Nicolas Mathevon. ✉email: mathevon@univ-st-etienne.fr

Crying during infancy is an innate survival mechanism that evolved in humans and many other animals to maintain proximity to and obtain care from caregivers[1–5]. Commonly triggered by isolation, hunger, discomfort or pain, human infant cries are characterized by specialized acoustic features such as severe roughness under extreme distress[6]. Acoustic features of infant distress vocalizations share similarities across terrestrial mammals, from rodents, dogs and deer to primates, including humans[2]. This shared acoustic structure almost certainly arose from the shared adaptive function of a cry: exploiting the hearing sensitivities of listeners to elicit aid[7–9]. Indeed, human baby cries consistently and cross-culturally lead to interventions such as holding or feeding that can be critical to an infant's welfare[10,11].

Despite the universality and tremendous biological relevance of the human infant cry, there remains little consensus on what information cries can convey[4]. For example, the remarkably common question "why is my baby crying?" returns hundreds of thousands of contradictory results from online search engines. To the same question, the *Web of Science* returns a large number of scientific articles and books dating back more than a century. Despite this enormous interest expressed by the general public and scientific community, it remains unclear whether human baby cries encode their ostensible cause, and whether adult listeners can correctly decipher this cause. While such a capacity could benefit both the caregiver and the infant[4], it may be trumped by selection pressure to preserve stable vocal indices of identity in babies' cries that likely increased infants' chances of survival throughout human history[12].

Traditionally, infant cries were thought to represent acoustically distinct cry types, each associated with a discrete cause such as birth cries, pain cries, hunger cries, pleasure cries, startle cries, and attention cries[13,14]. Early studies suggested that mothers and trained nurses could identify such cry types without additional contextual cues[14,15]. However, recent research suggests that identifying discrete information related to the cause of a cry is not so straightforward. For example, while parents can discriminate between highly distinct cry contexts, such as pain versus mild discomfort[6], discrimination is substantially degraded for cries that share a similar level of distress[16]. Moreover, it has been shown experimentally that an adult listener's ability to classify a cry as communicating pain or discomfort is highly dependent on their prior experience with infants[17].

Although these previous studies suggest that it is difficult to identify the cause of a cry by ear, it is commonly believed, especially by the general public, that parents can discern why their baby is crying just by listening. Books on parenting echo this belief, while countless websites offer recipes to decipher babies' cries. Some non-academic sources even suggest that babies' cries are a "language" made up of phonemes whose meaning can be learned, and mobile applications proposing to decode babies' cries are becoming increasingly popular, despite a lack of fundamental scientific evidence to support their veracity. Moreover, researchers have yet to test the hypothesis that each baby may develop his or her own coding strategy for the cause of crying, as we test here. The question of whether baby cries encode information about their cause has therefore not yet been effectively answered.

Besides experiments testing whether human listeners can decode baby cries, recent attempts to develop automated methods to detect context-specific cry types, including pathological cries for clinical applications, have also produced mixed results[18–20]. A number of studies using various methods of acoustic analysis followed by machine learning have shown that it is possible to distinguish cries expressing strong pain from cries due to another cause. For example, cries were categorized with 71.68% reliability by analyzing visual features extracted from spectrograms of cries caused by pain versus non-pain stimuli[21–23]. In contrast, few studies have tested whether crying can categorically encode the most common types of distress experienced by infants such as hunger, separation, or simple discomfort. A major reason for this is the difficulty of obtaining databases with a sufficient number of well-documented cry recordings in these everyday contexts, for babies of a given age, and with enough replicates per baby[20]. Most studies testing whether cries carry information about their cause have therefore been conducted on a limited number of cry databases (see[24] for an overview). Problematically, most of the used datasets include a mixture of cries from healthy and sick or disabled babies, or babies with a high pathological risk. For example, The *Baby Chillanto* database[24] lists five types of cries including "deaf", "asphyxia", "normal", "hungry", and "pain".

Few cry databases are focused solely on healthy babies. One exception, the *Donate a Cry* dataset[25], is still relatively undocumented[20]. Using only 150 baby cries from this database, a previous study[26] classified five cry types ("hunger", "attention seeking", "lack of ease", "stomach issue", "unidentified reasons") with 81.27 % reliability. However, in addition to the very small sample of cries, the fact that the babies included in the database ranged between 0 and 2 years of age makes the result of the study difficult to interpret, as age may explain substantial variance in cry acoustics. A similar bias is found in another study[27] where the authors, despite having obtained good classification scores, also mixed cries from babies of very different ages (1–22 months; 320 cries extracted). The *Dunstan Baby* database[28] is another database of cries traditionally used to test cry classification methods, and was established for commercial purposes. The database contains a limited number of cries recorded at different ages (between 1 day and 6 months), classified into five categories defined without scientific validation (« Neh » hungry, « Eh » Pain/burp-me, « Owh » sleepy, « Eairh » Pain, and « Heh » discomfort). After a meticulous review of the cries available in this database, some authors[29] were able to extract only 400 s of usable audio recordings (83 cries from 39 babies; see also[30]). They obtained excellent recognition scores for all five cry causes. As the authors point out, however, the small number of cry recordings they used increases the risk of overfitting in the learning models. Using the same database but different analysis methods, other studies[31,32] obtained comparable results. However, the problem of mixing cries from babies of different ages, and the fact that it is probably not the same babies who were recorded longitudinally, makes the interpretation of these results very difficult. Another recurrent problem with available databases of infant cries is that the identity of the babies is often not known. It thus cannot be excluded that some cries in certain categories may have been produced by the same babies, leading to pseudo-replication issues and falsely inflating recognition accuracy.

Taken together, the search for automatic methods to identify the causes of baby cries cannot be considered to have been successful to date. This is not because acoustic analysis and machine learning methods are inefficient—on the contrary, they are now particularly elaborate. Rather, a key problem is that modern analytical methods have not been previously applied to a large and well-controlled corpora of cry recordings. Here, we present a dataset containing cries from a cohort of babies recorded systematically and longitudinally at multiple ages during the first months of life and whose causes of crying have been labeled in a bottom-up, systematic manner. With this material, we can test whether a baby develops its own way of encoding information about the cause of crying.

A parallel body of literature describes the human infant cry as largely divorced from context[33]. Instead, some researchers argue, variations in cry acoustics reflect the baby's level of distress regardless of cry cause. According to this "graded cry" hypothesis,

more extreme distress levels predict increasingly severe cry acoustics, as observed in other primate species[8]. For example, human infant cries are often characterized by nonlinear phenomena that arise from aperiodic vibration of the vocal folds and that contribute to the perceived roughness of vocalizations[34,35]. Such roughness increases with distress, for example in cries produced during a vaccination compared to during a bath[6]. Infant cries also show an acute spike in their fundamental frequency ($f_o$, perceived as pitch) when distress levels reach an upper threshold[36], likely due to excessive vocal fold tension. Perception experiments show that adult listeners associate these acoustic variations with the ostensible level of distress or pain experienced by the crying infant[6,16,37], offering converging support for the graded cry hypothesis.

The ongoing debate surrounding whether human infant cries dynamically encode discrete causes is further complicated by salient individual differences in cries across babies, which researchers have often failed to control for in acoustic analyses (see[38] for discussion). Indeed, cries have long been known to differ acoustically from one baby to another[12,38], and we have experimental evidence of the ability of parents and naive listeners to recognize specific infant calls very reliably[39,40]. This is why it is critical to control for baby identity when testing whether baby cries communicate their cause. Individual vocal signatures in babies' cries have most consistently been tied to individual differences in $f_o$ (pitch)[41] and are already present by 3 months of age[42]. Remarkably, individual differences in $f_o$ remain stable across the human lifespan, from infancy to childhood[43] and from childhood to adulthood[44], suggesting that the human voice may function as a stable biomarker of identity, beginning already at birth. If baby cries have been shaped by selection to provide reliable indexical information about the infant, such as their sex, age, and identity, we can expect some degree of stability and predictability in cry acoustics within individual babies.

Although $f_o$ is known to differ across babies[42], standardized fine-grained acoustic analyses of cry signatures are lacking[1,4]. It thus remains unknown how much variance in cry acoustics can be explained by static traits such as sex and age. Because sexual dimorphism of the vocal anatomy does not emerge until puberty in humans[45], we do not expect sex differences to be present in baby cry acoustics. Indeed, in a previous study, we found no difference between the pitch of female and male baby cries[42]. A previous study[46] also points out that sex does not directly predict variance in cries, though peripheral estradiol concentrations (a baby-specific characteristic) do predict infant vocal performance. In the present study, we did not investigate hormone levels, but we did test for effects of sex and age. While we did not predict sex differences, we do expect that babies' cries will evolve with age. Indeed, expansion of the vocal apparatus, developmental maturation of neurocognitive capacities, and progressive changes in parent-infant communication strategies may all contribute to age-related vocal changes in the first months of life[5]. At the same time, anatomical constraints and selection pressure for indexical signaling may lead to stability in individual cry signatures during early ontogeny, for example, to facilitate parent-infant recognition, bonding, and to allow parents to familiarize themselves with the cries of their infant allowing them to extract dynamic cues to cry context or urgency[4,39]. Evidence for within-baby stability in cry signatures would also raise the possibility that contextual cues in infant cries may be specific to each infant, a hypothesis that has yet to be tested.

In this longitudinal study, we test for acoustic variability and stability in human baby cries, both across infants and across the first 4 months of each infant's life. We test the predictions that babies have individual cry signatures and that these individual cry signatures do not differ between the sexes and remain stable with age. We also test the prediction that there are consistent acoustic differences between baby cries caused by different events. To achieve these objectives, we audio recorded 24 male and female babies during several continuous 48-h sessions in their homes at 15 days, 1.5 months, 2.5 months, and 3.5 months after birth. We classified cries into three key contexts (hunger, isolation, or discomfort), relying mainly on the parental behavior that ceased the cry, which also corresponded to the parental evaluation of cry cause in 75% of cases. Crying bouts were segmented into 39201 individual cries, from which we measured ecologically relevant acoustic parameters including fundamental frequency (perceived as pitch), perturbation and noise parameters, roughness, and duration. These cries and their associated metadata have been collated into a database available online: *EnesBabyCries1* (see Methods and Supplementary Data). Using mixed models and machine learning, we tested whether these acoustic characteristics vary as a function of cry cause and whether this information is universally coded or specific to each baby. To verify the robustness of the results we derived from our acoustic analyses, nearly 250 male and female adult listeners judged the causes of these cries in two psychoacoustic perception experiments involving either an implicit or explicit training phase.

Combining longitudinal naturalistic recordings of human baby cries across contexts, fine-grained acoustic analysis, machine learning, and psychoacoustic experiments with human listeners, we answer long-standing questions about the encoding and decoding of evolutionarily relevant information in human infant cries: Are infant cries sexually dimorphic? Which acoustic parameters contribute to individual cry signatures? Are cry signatures stable across the first 4 months of life? Does the acoustic structure of baby cries differ by cause, either universally or within infants? And finally, to what extent can adult listeners assess the cause of babies' cries when trained on a specific infant?

## Methods
The research was performed under the authorizations 18CH085, no.IDRCB 2018-A01399-46 (recordings of cries) and IRBN692019/CHUSTE (psycho-acoustic experiments) and approved by the French national human ethics committee: *Comité de Protection des Personnes*. The study, including sample characteristics, design, procedures, and outcome measures was preregistered on https://clinicaltrials.gov/ under no. NCT03716882. The preregistration did not specify the hypotheses or the full analysis plan. Informed consent was obtained from all parents. Parents consented to public sharing of the cry-data. The playback experiment was approved by the local ethics committee (Comité d'Ethique du CHU de Saint-Etienne, Institutional Review Board: IORG0007394), and informed consent was obtained from all participants.

**Acoustic recording**. To obtain recordings of infant cries in real-life contexts, we selected 30 families recruited during their stay in the maternity ward of the University Hospital of Saint Etienne and living in the vicinity of Saint-Etienne, France. Mothers were visited by a pediatric resident (MB) during working hours on one of the 3 days of their stay in the maternity ward and invited to take part in the research project. Key inclusion criteria included full-term childbirth and eutrophy. We therefore did not include babies with intrauterine growth retardation, antenatal or established neurological pathologies at birth, perinatal asphyxia, encephalopathy, known antenatal pathology or those born from a multiple pregnancy. As five families withdrew from the study and one more baby was not recorded due to technical issues, our final sample included 24 babies (10 girls and 14 boys) from 24 families (Table 1). The sex of the babies was given by the parents and was

**Table 1 Biological data of 24 recorded babies.**

| Baby ID | Birthdate (dd/mm/yy) | Sex (M/F) | GA (weeks) | DW (CS/V) | Apgar at 5 min | BW (g) | BH (cm) | BHC (cm) | Deafness Screening results | Pg | N B | F-Ch |
|---|---|---|---|---|---|---|---|---|---|---|---|---|
| RB01 | 12/10/18 | M | 38 | CS | 10 | 3615 | 51 | 35 | NL | 1 | 1 | 1 |
| KA02 | 16/10/18 | M | 39 | CS | 10 | 3805 | 52 | 37 | NL | 1 | 1 | 1 |
| BS03 | 16/10/18 | F | 38 | CS | 10 | 3415 | 50 | 35 | NL | 3 | 2 | 2 |
| TM04 | 19/10/18 | M | 37 | V | 10 | 2135 | 42 | 32 | NL | 2 | 2 | 2 |
| TA05 | 23/10/18 | F | 40 | V | 10 | 3395 | 50 | 33 | NL | 1 | 1 | 1 |
| BR07 | 26/10/18 | M | 39 | CS | 10 | 3710 | 51 | 35.5 | NL | 2 | 2 | 2 |
| PA08 | 06/11/18 | M | 38 | V | 10 | 2710 | 48.5 | 33 | NL | 2 | 2 | 2 |
| LC10 | 11/11/18 | F | 39 | V | 10 | 3245 | 50 | 33 | NL | 1 | 1 | 1 |
| CA12 | 01/01/19 | F | 37 | V | 10 | 2995 | 49 | 34.5 | NL | 2 | 2 | 2 |
| GL13 | 06/01/19 | M | 40 | V | 10 | 3865 | 48.5 | 35 | NL | 1 | 1 | 1 |
| PJ14 | 06/01/19 | F | 41 | V | 10 | 3510 | 51 | 35.5 | NL | 2 | 1 | 1 |
| BR15 | 14/01/19 | F | 39 | V | 10 | 2910 | 47 | 33 | NL | 1 | 1 | 1 |
| TL16 | 07/01/19 | M | 40 | V | 10 | 3170 | 50.5 | 31 | NL | 1 | 1 | 1 |
| XM17 | 20/01/19 | F | 40 | V | 10 | 3895 | 51 | 34 | NL | 1 | 1 | 1 |
| PE18 | 22/12/18 | M | 38 | V | 10 | 3235 | 49 | 35.5 | NL | 5 | 3 | 3 |
| SA20 | 29/01/19 | M | 40 | V | 10 | 3580 | 50.5 | 34.5 | NL | 2 | 2 | 2 |
| ML21 | 26/02/19 | M | 41 | V | 10 | 3600 | 52.5 | 33.5 | NL | 2 | 1 | 1 |
| HA22 | 28/02/19 | F | 39 | V | 10 | 3285 | MD | 35 | NL | 1 | 1 | 1 |
| SB23 | 27/02/19 | M | 39 | V | 10 | 2675 | 47 | 33 | NL | 2 | 2 | 2 |
| PB24 | 02/03/19 | M | 38 | V | 10 | 2790 | 47 | 32 | NL | 1 | 1 | 1 |
| LC26 | 03/04/19 | M | 41 | V | 10 | 3595 | 51 | 35 | NL | 2 | 2 | 2 |
| PA27 | 03/04/19 | F | 39 | V | 10 | 3100 | 50 | 35 | NL | 2 | 2 | 2 |
| BM29 | 31/08/19 | M | 39 | V | 10 | 3900 | 52 | 35 | NL | 1 | 1 | 1 |
| MR30 | 23/10/19 | F | 39 | V | 10 | 3265 | 48 | 35.5 | NL | 1 | 1 | 1 |

*dd* day, *mm* month, *yy* year, *M* male (boy), *F* female (girl), *GA* gestational age, *DW* delivery way, *CS* cesarean section, *V* vaginal delivery, *BW* birth weight, *BH* birth height, *MD* missing data, *BHC* birth head circumference, *NL* normal, *Pg* number of pregnancies for the mother, *B* number of birth for the mother, *F-Ch* number of children for the father.

consistent with their civil status. We collected no data on race nor ethnicity.

Recordings were conducted inside each family's private home during several 48-h sessions at 15 days, 1.5 months, 2.5 months, and 3.5 months following birth. Because of variations in parental availability and the well-known decrease in the frequency of infant crying with age[1], some babies could not be recorded at each of the four ages (Supplementary Table 1). We thus obtained recordings from 17 infants at 15 days, 24 infants at 1.5 and 2.5 months, and 12 infants at 3.5 months.

We used automated sound recorders equipped with omnidirectional microphones (Song-Meter SM4 Acoustic Recorder©, Wildlife Acoustics, Inc., Concord, MA, USA). The acoustic recorder was placed in the baby's room at a height of 1.2–2.1 meters and at a distance of 1–4 meters from the baby to ensure a high signal-to-noise ratio. Recordings were made at 44.1 kHz with a 16-bit resolution and saved as WAV files. In total, we obtained around 3600 h of audio recording (24 babies * 4 recording sessions—21 missing recording sessions * 48 h).

**Parental questionnaire on cry causes.** During each 48-h recording period, parents completed a form indicating, for each crying sequence produced by their baby, the onset time of the cry, the potential cause identified for the cry, the action(s) taken to stop the cry, and the action that was ultimately effective. Parents chose from the following causes: hunger, isolation, physical discomfort (such as fever, cold temperature, full diaper), pain, and unknown. There was also an option for parents to indicate a cry cause that was not listed via an open response comment box. In 75% of the cases, the action that stopped the crying matched the parental assessment of the cry cause (for example, parents indicated "hunger," and feeding their child did indeed stop the cry). When the parent's assumed cause did not match the action that stopped the cry, we coded the cause of the cry based on that

action and not on the parental assessment. We made this choice to increase the reliability and objectivity of labeling cry causes.

**Extraction of cries.** Cries were selected and edited following a six-step process (see Fig. 1). In step 1, we manually extracted audio cry sequences from each 48-h recording session using Praat software version 6.1.16[47]. Each cry sequence corresponded to a single questionnaire entry completed by the parents, and thus to a single crying bout and cause. We obtained 3308 cry sequences, which were then classified by their acoustic quality in step 2 as: (1) excellent signals with no interfering environmental noise; (2) acceptable signals with only short durations of interfering noise; and (3) highly noisy signals. Noise was identified as any overlapping background sound, corresponding in most cases to parental voices or distant sounds, for example from televisions or music. Only sequences classified as excellent or acceptable were retained for subsequent editing (total: 676 cry sequences from 24 babies; average sequence duration 49 ± 74 s, range 1–73 s). In step 3, any cry sequences with short durations of noise were cleaned by manually cutting out those bits of background noise, resulting in clean, spliced cry sequences.

In step 4, to standardize cry signals for acoustical analysis, the 676 clean cry sequences were automatically segmented into 78094 vocalizations with the *segment* function in *soundgen* 2.0.0[48]. We looked for segments that were a minimum 50 ms in duration, separated from other cries by at least 100 ms. More details on the segmentation algorithm are available in the documentation of the *soundgen* function *segment*, and the R code for segmentation is provided in the supplements (scripts.zip, file prep.R).

In step 5, we removed all non-cry vocalizations produced by infants (e.g., unvoiced grunts, coughing), thus retaining only cries, operationalized as at least partly voiced episodes of vocalizing that were different from steady background noise and that satisfied the following four conditions: (1) minimum 20% voiced; (2) median pitch > 150 Hz; (3) duration > 250 ms;

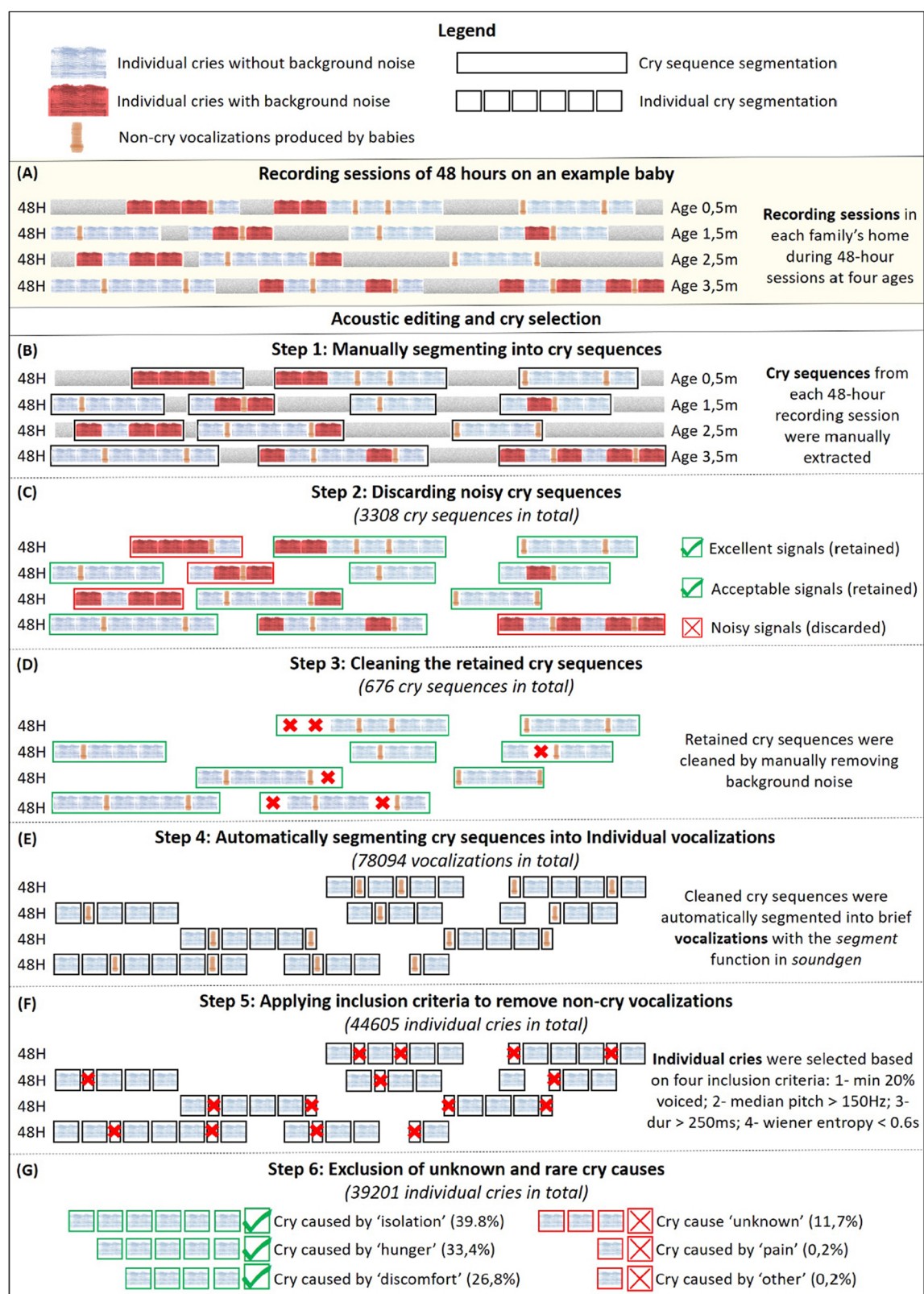

and (4) Wiener entropy < 0.6. These inclusion criteria yielded the lowest rate of false positives (2% of non-cry vocalizations) in a randomly drawn sample of 100 selected and 100 rejected cry syllables. The automatic segmentation resulted in a total of 44605 cries across infants and ages.

Finally, in step 6, we removed cries classified by parents as having been caused by pain (95 cries) and those classified as "other" (e.g., fear, fatigue, colic, excessive noises, awakened; 76 cries), as these responses each represented less than 0.2% of all cries. We also removed cries labeled with an "unknown" cause (11.7% of cries). The three major cry causes (isolation, hunger, and discomfort) were thus represented by 15609 (39.8%), 13095 (33.4%), and 10497 (26.8%) cries, respectively, with an average duration of 860 ± 590 ms (range 0.220–12 s). The final sample for

**Fig. 1 The six steps of the cry selection and editing process. A** Recordings were conducted inside each family's private home during several 48-h sessions at 15 days, 1.5 months, 2.5 months, and 3.5 months following birth. Cries were then selected and edited following a six-step process: (**B**) step 1: manual extraction of audio cry sequences; (**C**) step 2: only sequences classified as excellent (no interfering environmental noise) or acceptable (only short durations of interfering noise) were retained; (**D**) step 3: retained sequences were cleaned by manually cutting out the remaining bits of background noise; (**E**) step 4: Automatic segmentation, resulting in cry segments of 50 ms minimal duration separated from other cries by at least 100 ms; (**F**) step 5: Removal of non-cry vocalizations; (**G**): step 6: removal of cries classified by parents as having been caused by pain and those classified as "other" (less than 0.2% of all cries). Only recordings obtained in step 6 were used for the present study. The *EnesBabyCries1* database contains the recordings after step 3 (*EnesBabyCries1A*) and after step 5 (*EnesBabyCries1B*). The database is available online at https://osf.io/ru7na/ (see Supplementary Data for a description of the database).

acoustic and statistical analysis included 39201 cries (Supplementary Table 1).

We used short cries as our unit of acoustic and statistical analysis, justified by two important factors. First, recordings were taken in real-life environments, wherein background noise was inevitable and needed to be removed to ensure high-quality audio for robust acoustic analysis. Cry sequences were thus manually cut to remove background noise (steps 2–3), resulting in spliced cry sequences. Second, by measuring acoustic parameters over an entire sequence, we would have introduced a greater degree of acoustic variation, making it difficult to assess the contributions of individual acoustic parameters.

The cries collected for this study are now available in the form of a new cry databank (*EnesBabyCries1*, Supplementary Data). This bank contains two sets of recordings (*EnesBabyCries1A* and *EnesBabyCries1B*). *EnesBabyCries1A* contains the sequences of recordings from step 3 (i.e., after removal of noisy parts). *EnesBabyCries1B* contains the cries from step 6 (after segmentation and selection of the cries; these cries are the ones used in the present research). This databank is anonymized. No clues to the identity of the baby have been retained in the recordings. The metadata accompanying the cries are: (1) the age of the recorded baby; (2) the baby's biological sex; (3) the cause of the cry as stated by the parent; (4) the parental action that ended the cry. We obtained parental permission to make these recordings and information public. The databank *EnesBabyCries1* is available here: https://osf.io/ru7na/.

**Acoustic analysis**. The acoustic structure of each of the 39201 cries was represented by ten key acoustic variables. The choice of these variables was based on a large body of research in animal communication and the human voice sciences, implicating these acoustic variables as markers to speaker identity, speaker physical traits, and/or motivation and emotion[9,49–51]. A key acoustic parameter in human nonverbal vocal communication is fundamental frequency, perceived as voice pitch, which is highly individual and stable between individuals ([44]), and yet also dynamic within individuals and thus critical in the communication of effect and emotion[52]. We hence included the following acoustic parameters: *Median Pitch* (median fundamental frequency $f_o$, given in hertz) as a measure of central tendency that is more robust to noise in pitch tracking than is the mean pitch and *Pitch IQR* (interquartile range of $f_o$, in Hz), which was used instead of standard deviation or range as such measures are less sensitive to noise (i.e., incorrectly measured pitch in some voiced frames). The overall proportion of frames that are voiced was also included as it can distinguish between mostly tonal whine-like cries and wheezy or breathy vocalizations, but also because measures of voice quality were calculated specifically for voiced frames (*Voicing*, scaling from 0 to 1). Voice quality or "timbre", understood as any acoustic characteristics that distinguish between two voices at the same intensity and pitch, was captured by several acoustic variables including *Spectral centroid* (median spectral centre of gravity of voiced segments, in Hz), which

indicates how much energy is present in high versus low frequencies and distinguishes between bright or shrill and relatively "dark" voices. Finally, several acoustic parameters were measured to describe tonal versus noisy vocalisations: *Entropy* (median Wiener entropy, scaling from 0 to 1), *Harmonics-to-noise ratio* (measure of harmonicity, in decibels), *Jitter* (short-term disturbances in $f_o$, given as a percentage), *Shimmer* (short-term disturbances in the amplitude of the sound signal, given as a percentage), and *Roughness* (median proportion of modulation spectrum of voiced segments within the roughness range of amplitude modulation 30–150 Hz, given as a percentage). Working with short cries, temporal structure cannot be captured, apart from one obvious descriptive—cry *Duration* (in seconds). Amplitude (loudness) of cries could not be computed because the recording distance (microphone to baby) was not perfectly standardized. Acoustic measurements were performed in *soundgen* 2.0.0[48], except for *jitter* and *shimmer*, which were measured in Praat[47].

**Statistical analysis of cry features**. Each acoustic variable was normalized to a mean of 0 and SD of 1 within each baby. The acoustic differences between cries according to their apparent cause (hunger, discomfort, or isolation) and baby age were assessed with multivariate Bayesian mixed models fit in R with the *brms* package, version 2.17.0[53]. Because multiple vocalizations were extracted from the same sequences of crying, we also included a random intercept per crying sequence, as well as per baby. The model was: *mvbind(duration, entropy, HNR, jitter, pitch_iqr, pitch_median, roughness, shimmer, spectralCentroid, voiced) ~cause \*age + (cause \* age|baby) + (1|sequence).*

We used a Random Forest classifier, *randomForest* package in R, version 4.7-1[54], with the same 10 acoustic predictors and with stratification per cause (i.e., without over-representing more common causes) to predict cry cause from acoustics using machine learning. Two-thirds of the available observations were used for training, and one-third for testing. Crucially, we trained our models on cries from one set of recording sessions and tested them not only on different cries but also on cries taken from different recording sessions. Because cries from the same recording session may share not only the same internal baby's state but also background noises and other confounds, training, and testing on different sessions ensures that the model learns the categories of interest (causes of crying, babies' identities, etc.) rather than session-specific acoustic signatures. The Random Forest algorithm is stochastic, and the training sample was chosen at random; therefore, we ran each model 100 times with different training and test sets and summarized its performance by median accuracy and 95% coverage intervals.

Considering the large dataset of cries (39201 cries), it was essential to develop an effective method to visualize similarities across cries based on various categories such as individual babies, age groups, or cry causes. The crux of this problem is obtaining a reliable and intuitive measure of similarity across cries. To solve this issue, we present a method based on dynamic time warping

(DTW) of frame-by-frame acoustic features (fundamental frequency, harmonicity, etc.). To reduce the dimensionality of the resulting distance matrix, we use the state-of-the-art algorithm of Uniform Manifold Approximation and Projection (UMAP) implemented in R with the *uwot* package version 0.1.11[55]. UMAP is conceptually similar to traditional unsupervised methods for dimensionality reduction, such as principal components, multidimensional scaling, and more recent techniques like tSNE, but UMAP uses rigorous mathematical methods of topological analysis, preserves global structure and within-cluster distances, and scales well to very large datasets, as recently demonstrated in complex comparative analyses of animal vocal acoustics[56]. Importantly for our purposes, UMAP accepts a pre-calculated distance matrix as input, which in this case was the output of dynamic time warp; calculating this matrix only once is much faster than using DTW as a distance metric inside UMAP, making it possible to analyze tens of thousands of sounds without the need for cluster computing.

**Playback experiments**. Two independent samples of 146 adult listeners (36 mothers, 37 fathers, 38 non-mother women, 35 non-father men; mean age ± sd = 28.0 ± 6.4 years, range 18–40 years) and 102 adult listeners (26 mothers, 25 fathers, 24 non-mother women, 27 non-father men; mean age ± sd = 27.1 ± 5.6 years, range 18–40 years) took part in playback Experiments 1 and 2, respectively. Participants were recruited on the online recruitment platform *Prolific*[57], where they were redirected to the online testing platform *Labvanced*[58] on which the experiments were designed and hosted. Participants were paid for their time at the recommended rate of 7.5 GBP per hour.

For these perception experiments, we selected cries from 1.5-month-old babies who produced at least 36 cries longer than 0.7 s for each context. By eliminating shorter cries, we aimed to ensure that listeners have sufficient acoustic information from each cry. We focused on a single age group because, as our analyses show, individual cry signatures change with age. Only seven male infants and one female infant met these criteria and thus, to avoid possible confounding effects of biological sex, we used only the cries of male infants as playback stimuli (total of 2430 cries; mean duration ± sd = 1.34 ± 0.51 s, range [0.80, 5.70]). For implicit training in Experiment 1, we prepared two series of 12 cries for each cry cause (hunger, discomfort, isolation), resulting in six sets per participant, with all cries emitted by the same baby. For explicit training in Experiment 2, we prepared one set of 69 cries, with 23 cries per cause. For the testing phase of both experiments, we prepared one set of 30 cries, with 10 cries from each cry cause. Training and testing sessions involved cries from the same baby, with different cries in the training versus testing sessions, in order to maximize the ecological validity of our results. Stimuli were fully randomized within sessions. All listeners completed a short questionnaire indicating their age, gender, and whether they were biological parents. To control for potential experience with babies' cries among nonparents, all participants reported their experience with infants, including whether and how often they currently have contact with and/or have cared for babies aged less than 2-years.

Both experiments consisted of a training phase and a testing phase. In Experiment 1, listeners were implicitly familiarized with the cries of a single baby. They assessed the level of either hunger, discomfort or isolation of the baby from its cry using a sliding scale ranging from *Not at all* to *Extremely*. Examples of discomfort included cold, fever, and full diaper, whereas isolation was defined as the infant seeking contact or attention. This task ensured that listeners were actively attending to cries for maximal familiarization via implicit training. In the subsequent test

session, listeners indicated whether each cry was due to hunger, discomfort, or isolation in a three-alternative forced-choice task. In Experiment 2, listeners were explicitly trained on the cause of a subset of cries. For each cry in this training phase, listeners indicated if the cry was due to hunger, discomfort, or isolation in a three-alternative forced-choice task, after which they received explicit feedback regarding the actual cause of the cry. They were informed that their goal was to use this explicit feedback to learn to recognize the cause of baby cries in a forthcoming test. The final 30 trials, which included cries the participants had not heard during the training phase, were considered the testing phase.

Unaggregated, trial-by-trial responses in both psychoacoustic experiments were analyzed using Bayesian mixed models fit with the *brms* package[53]. Cry cause, number of previous exposures to cries from the same recording session (*nSameSession*), participant sex, and parental status (parent or nonparent) were included as fixed factors, while participant and baby identity were included as random factors. Fixed factor interactions and random slopes were defined using *WAIC*, Watanabe-Akaike information criterion[59] for model selection. The model structure was as follows:

$$success \sim nSameSession + sex + cause + parentality$$
$$+ \, cause : parentality + sex : parentality$$
$$+ \, (1|subjectID + babyID)$$

**Reporting summary**. Further information on research design is available in the Nature Portfolio Reporting Summary linked to this article.

## Results

**Acoustic encoding of sex and age**. We first tested whether individual cries, regardless of cry context, carry acoustic information about the vocalizing infant's sex and age based on ten acoustic predictors known to be important in human and animal nonverbal communication: average fundamental frequency (median pitch) and its interquartile range (pitch IQR), entropy, roughness, spectral centroid, harmonicity, jitter, shimmer, duration, and voicing (see Methods for a detailed description of the measured acoustic variables and their justification; see also[49–51]). Together, these acoustic parameters capture the most biologically and perceptually relevant cry characteristics including voice pitch and its variability, voice quality, and duration.

Multivariate Bayesian mixed models did not reveal any consistent acoustic differences between the cries of infant boys and girls, either overall or for any specific age group (Fig. 2a), corroborating previous work on 3-month-old babies[42]. While infant cries did not differ between sexes, we found consistent acoustic changes in cries as babies grew older (Fig. 2b). Notably, cries became considerably more tonal with age, as evidenced by an increase in the harmonics-to-noise ratio (average increase per month of age: 0.17 SD, 95% CI [0.05, 0.29]) and a very rapid drop in the entropy of cries ($-0.42$ SD per month $[-0.55, -0.29]$). With each additional month of age, we also observed an increase in the proportion of voiced frames (0.24 [0.12, 0.36]) and a slight increase in cry fundamental frequency (herein, pitch) (0.11 [0.04, 0.17]), whereas spectral centroid became lower with each month of ageing ($-0.19$ $[-0.34, -0.05]$).

Taken together, we show that human infants gradually shift from producing mainly noisy and shrill cries to producing more tonal and melodious cries from birth to nearly 4-months of age (Fig. 2b). However, accurate estimation of actual age from a single cry of any given baby was low. Indeed, a Random Forest classifier (randomForest R package[54]; see Methods), using the same ten acoustic predictors, could recognize a baby's age group with a moderate accuracy of about 40% [35, 43], with an Odds Ratio to

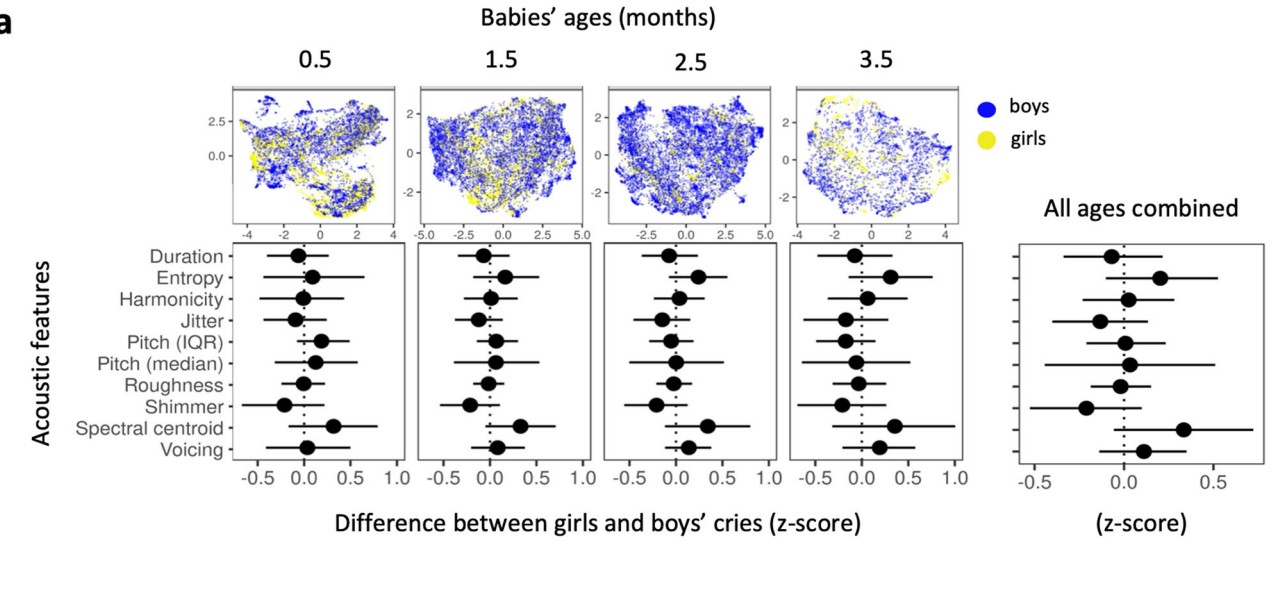

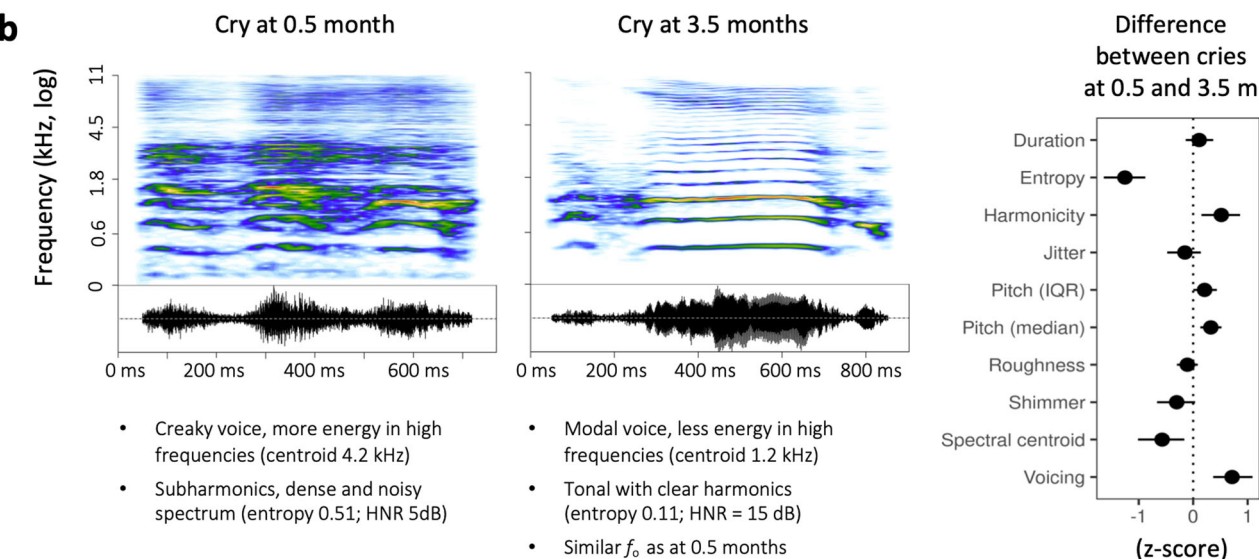

**Fig. 2 Sex and age information in human baby cries. a** Absence of a sex signature. Top: The distribution of baby boys' and girls' cries ($N = 39201$) in two-dimensional UMAP acoustic spaces shows strong acoustic similarities between the cries of both sexes, emphasizing the lack of sex differences in human baby cries (each dot represents one cry, acoustic spaces are obtained using UMAP, see Methods). Bottom: No acoustic descriptor varied between sexes, either for each age independently or for all ages combined (multivariate Bayesian mixed models, medians of posterior distributions with 95% CI). **b** Cry acoustics change systematically with age in the first 4 months of life. Left: Example spectrograms of two cries recorded from the same baby boy at 0.5 and 3.5 months of age, illustrating changes in key acoustic characteristics over this time (50 ms Gaussian window, frequency in kHz on a bark-spaced scale). Right: Changes in acoustic predictors from 0.5 to 3.5 months of age, showing that acoustic indices of roughness decreased with age, while those indicating tonality increased (multivariate Bayesian mixed models, medians of posterior distributions with 95% CI; $N = 24$ babies, 39201 cries).

chance of 2.0, 95% CI [1.6, 2.2] [odds for model 40/(100−40) = 0.66; odds for random guessing 25/(100−25) = 0.33; 0.66/0.33 = 2].

This lack of robust age signatures may be explained by strong inter-individual variation across infants. As a next step, we therefore precisely analyzed individual signatures in cries and their development in early ontogeny.

**Acoustic encoding of baby identity**. We observed individual cry signatures across infants. A Random Forest model using the same ten acoustic predictors but normalized across all observations instead of within each baby achieved an accuracy of ~28% (95% CI [23, 31]) when discriminating among all 24 babies from a single cry and across all ages. This is much better than chance (OR = 8.7 [6.8, 10.2]), and also constitutes a high rate of recognition from an ecological perspective, wherein infant recognition in real life would almost never involve having to identify one newborn baby from among two dozen others. Moreover, this is also a very conservative estimate as the model relies on only a fraction of the available acoustic predictors in infant cries, and cries from the same recording session never appeared in both training and test sets. All ten acoustic characteristics noticeably contributed to individual recognition of babies by Random Forest models, but median pitch was the top predictor. These results, therefore, support the hypothesis that babies' cries carry idiosyncratic acoustic characteristics that define an individual signature unique to each baby[12,38]. Further corroborating this result,

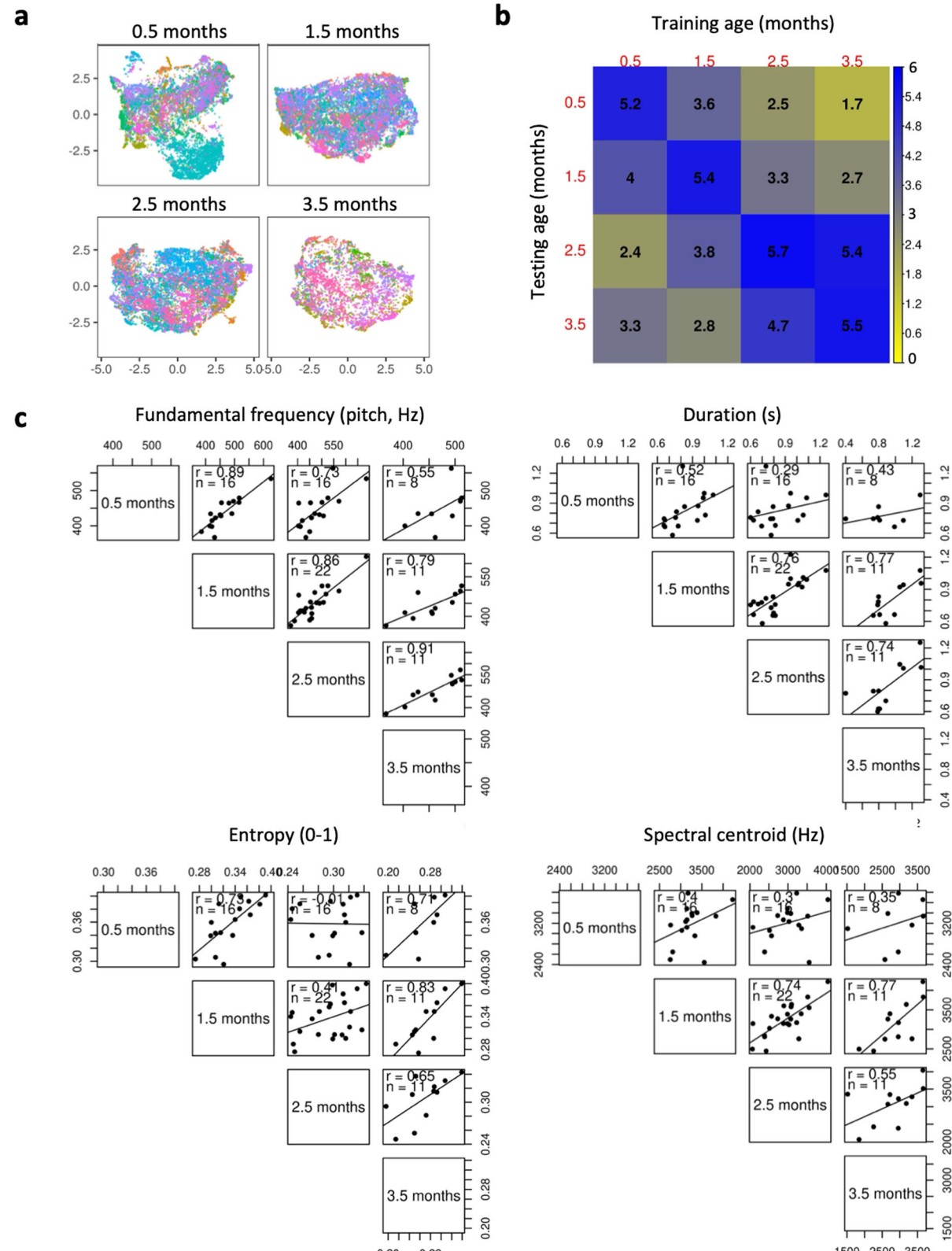

we observed some clustering by baby identity at each age in an unsupervised UMAP projection, a two-dimensional representation of the acoustic similarity between the analyzed cries (Fig. 3a).

Our results also show that individual signatures of human infants remain relatively stable throughout the first months of life. Indeed, models trained on cries recorded at a given age can still identify individual babies if tested with their cries recorded at other ages (Fig. 3b). Nevertheless, we observed developmental drift in cry signatures: the greater the difference in baby age between training and test datasets, the less reliable the model's recognition of infant identity (Fig. 3b). This indicates that cry signatures, while specific to each baby and relatively stable throughout early ontogeny, nevertheless change to some degree with age.

**Fig. 3 Individual vocal signatures in baby cries. a** Distributions in 2D acoustic spaces of baby cries with each color representing an individual baby identity suggest idiosyncratic vocal properties at each age ($N = 39201$ cries including 4580–12399 cries per age, 327–4140 cries per baby; acoustic spaces obtained with UMAP, see Methods). **b** Accuracy in classification of baby identity using a machine-learning classifier. The test-retest matrix shows the accuracy of classifying cries by baby identity with Random Forest models trained on cries from one age group and tested on either the same or a different age group (odds Ratio to chance; 1 = no better than chance), showing that recognition is highest in the same-age group, even though training and testing are performed on cries from different recording sessions. The dataset includes 15540 cries of six babies who were recorded at all four ages. Columns show the training-age group, and rows show the testing-age group. The greater the difference in age between training and testing, the less accurately the model can recognize the identity of a given baby. **c** Scatterplots showing the drift in individual babies' acoustic characteristics over the first 4 months of life. While acoustic characteristics change with age, they do so consistently for each baby (one data point = one baby). Not all babies were recorded at all four ages, so sample sizes vary across age groups, shown as $n$ within each scatterplot. The strength of pairwise relationships between the same vocal parameter at two different ages is shown in each scatterplot as the $r$ correlation coefficient.

This cry signature drift may be the result of how individual babies' voices change over time. However, because our results show that intra-individual cry acoustics remain stable throughout early development, we show that these ontogenetic changes largely follow a predictable trajectory for a given infant. For example, a baby whose cries are already high-pitched at 0.5 months of age relative to other babies retains relatively high-pitched cries at 3.5 months of age (Fig. 3c).

**Acoustic encoding of cry cause**. Finally, we tested whether the context in which cries were produced can be determined from the acoustic characteristics of cries. Our results show that hunger, discomfort, and isolation cries do not differ systematically in their acoustic structure and thus cannot be effectively segregated by cause when pooling the cries of all babies at all ages. Indeed, a Random Forest supervised classifier achieved an out-of-sample accuracy of only 36% (OR to chance = 1.1, 95% CI [1.0, 1.2]), indicating that there are no statistically robust acoustic differences among cries produced in each context. Using the cause of crying indicated by the parents in the questionnaire, instead of the action that stopped crying, produced nearly identical results with a classification accuracy of 35%, 95% CI [33, 36]. This is not meaningfully different from 36% [33, 38] based on the cause that stopped crying.

Including age group as a predictor in Random Forest classifiers, or training and testing the models within the same age class, failed to raise cry cause recognition accuracy above chance level (31% to 37%, OR to chance between 1.0 and 1.2 in each age group; Fig. 4a). Furthermore, after training Random Forest models on each age group and testing them on the same and other age groups, the accuracy of cry cause recognition did not improve with age and remained close to chance within each age group (Fig. 4b).

Our results corroborate previous findings that human babies have individual cry signatures. This raises the possibility that the cries of each baby may be characterized by a unique acoustic "code" linked to the specific cause of their cry. If so, we would predict more consistent acoustic differences between cries of hunger, discomfort, and isolation within each baby than across all babies. We did not find support for this. The recognition accuracy of a Random Forest model trained with the same ten acoustic predictors increased only slightly, from 36% [33, 38] to 43% [36, 50], after the inclusion of the baby's identity as another predictor of cry cause (OR boost from 1.1 [1.0, 1.2] to 1.5 [1.1, 2.0]). This small performance improvement might be partly attributed to differences in the distribution of causes of crying across babies. Therefore, we also trained a Random Forest classifier to recognize the cause of crying using cries from only a single baby, and then tested how well this model could detect the cause of crying either in the same baby or in all other remaining babies. Using ten acoustic parameters and age group as predictors, the average out-of-sample recognition accuracy of cry cause remained low at 38% [17, 65] (OR = 1.2 [0.4, 3.2]) for the same focal baby ($n = 22$

babies recorded in all contexts), similar to when classifying cries of non-focal babies (34% [30, 54], OR = 1.0 [0.9, 1.2]; Fig. 4c).

We also tested the possibility that several babies might share a similar acoustic strategy to code for cry cause, a strategy that may not be universal to the entire group, but rather shared by sub-groups of babies within the larger sample. To test this, we calculated distance matrices showing the acoustic similarity between all possible (276) pairs of 24 babies in each of three cry causes, using fitted values of baby-specific acoustic signatures from the Bayesian mixed model predicting acoustics from context (Fig. 4d). If sub-groups of babies share similar strategies, these distance matrices should be correlated, with specific baby pairs showing stronger acoustic similarities than other pairs across cry contexts. This was not the case: the strongest correlation between pairs of distance matrices was negligible ($r = .06$; Fig. 4d).

**Decoding of cry cause by human listeners**. Our acoustic modeling shows that baby cries do not vary across hunger, isolation, or discomfort contexts in a large pool of naturalistic cry recordings. However, the absence of evidence is not necessarily evidence of absence, and we thus set out to corroborate this null result by conducting two psychoacoustic perception experiments to test whether human listeners can gauge the causes of infant cries. In the first experiment, parents and nonparent men and women ($n = 146$, mean age $28 \pm 6.4$) were implicitly trained and familiarized with the cries of a specific infant by rating cry intensity, and then tasked with classifying new cries from the same infant by their cause (see Methods; to limit pseudo-replication, we used the cries of seven different babies; each participant was assigned one of these seven babies). The overall recognition accuracy of cry cause in a logistic multilevel model was 34.8% (95% CI [29.4, 40.2]), which is no different than chance. We found no differences in performance between parents and non-parents (−0.9% [−4.4, 2.7]) and little to no difference between men and women (3.6% [−0.2, 7.3]; Fig. 4e). In this experiment, nonparent men and women reported similar rates of prior experience with infants (women 39%, men 46%).

In the second experiment, a different sample of parents and nonparents ($n = 102$, mean age $27.1 \pm 5.6$) were explicitly trained with a cry classification task designed to mimic the Random Forest classifier used in our machine learning procedures. Listeners were again presented with cries from a single infant and tasked with classifying each cry by its ostensible cause. Unlike experiment 1, however, here listeners were provided with explicit feedback after each trial indicating the actual cause of the cry, before classifying new cries from the same infant by their cause (see Methods). The overall recognition accuracy of cry cause was again at chance level: 35.4% [31.7, 39.1]. While we found no effect of parental status (1.0% [−2.5, 4.6]), women performed slightly better than did men in identifying cry cause (4.0% [0.3, 7.6]) (Fig. 4e). Given that in this second experiment, more nonparent women reported prior experience with infants (66%) than did

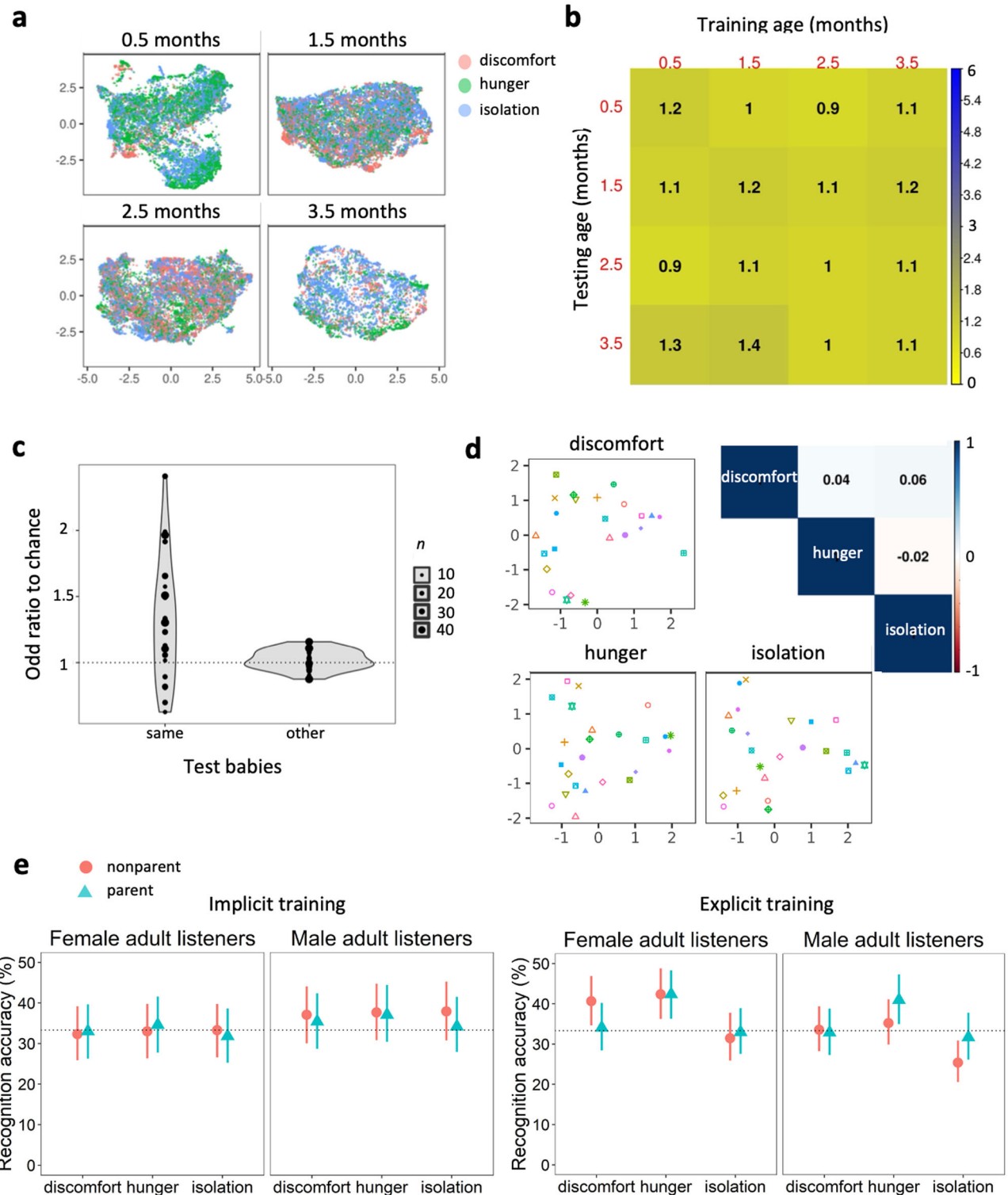

nonparent men (37%), this effect could be largely driven by sex differences in experience with babies' cries in this sample of listeners. Indeed, we found that while mothers performed no better than did fathers (1.3% [−3.8, 6.3]), nonparent women performed better than did nonparent men (6.7%, [1.7, 11.7]). Controlling for the number of previous exposures to cries from the same cry session had no apparent effect on recognition accuracy (odds ratio = 1.01 [1.00, 1.02]).

The results of our perception experiments thus show that, like machine learning algorithms, human listeners cannot consistently recognize the cause of crying from short recordings of human infant cries produced in the three most common contexts, regardless of their parental status or sex, and not even after brief training.

## Discussion

Our study demonstrates that baby cries are not sexually dimorphic, that they bear an individual signature established by an array of specific acoustic features that drift systematically with

**Fig. 4 Baby cries do not encode or communicate their cause. a** Projections of baby cries in two-dimensional acoustic spaces (UMAP) show poor clustering by crying context: discomfort ($n = 10497$ cries), hunger ($n = 13095$), and isolation ($n = 15609$). **b** Accuracy of context classification using a machine-learning classifier shows that cry cause recognition is around chance for all train-test combinations. The test-retest matrix shows the accuracy of classifying by context with Random Forest models trained on cries from one age group and tested on either the same or a different age group (odds Ratio to chance; $1 =$ no better than chance), for cries of six babies recorded at all four ages. **c** Accuracy of predicting the context of crying from acoustics with Random Forest models trained with the cries of one baby and tested on the same *versus* other babies. Dot size indicates the number of recording sessions in a model. For a few babies, cry cause is recognized above chance when a model is trained and tested on different sessions from the same baby, but with high uncertainty due to a limited amount of data per baby (best classification: OR $= 2.2$ [0.9, 4.8]). There is no transfer of information to non-focal babies. **d** Left and bottom: The acoustic distances between babies do not form clusters within each cry context, suggesting an absence of groups of babies sharing the same coding strategies. Top right: Distance matrices per context are not correlated. **e** Performance of adult listeners in classifying the cries of an assigned baby by context after an implicit training session (left panels, $n = 14892$ trials from 146 listeners) and after an explicit training session (right panels, $n = 10098$ trials from 102 listeners). Markers represent fitted values from a logistic mixed model (medians of posterior distribution and 95% CI; chance level 33.3%), where blue triangles indicate parents, and red circles indicate non-parents. To limit pseudo-replication, we used the cries of seven different babies. For each listener, we chose different cries for training and testing sessions.

age, and that they do not communicate robust information about their cause, at least not within our very large sample of brief individual cries. We show this using acoustic analyses and modeling of nearly 40,000 human baby cries recorded longitudinally along the first 4 months of life from 24 infants in distinct real-life contexts, paired with playback experiments on adult listeners that confirm an inability in both parents and nonparents to consistently identify the most common cause of cries produced by a familiar baby, even after training.

**No significant difference between boys' and girls' cries**. The absence of vocal cues to sex in the cries of babies confirms previous observations in older infants. Indeed, the pitch of cries in infants aged 3 months has been shown to be independent of their sex[42], in accordance with similarities in voice pitch between older boys and girls until puberty[60–63]. Here, we extend this observation over a span of 4 months in early ontogeny, from 15 days to 3.5 months postpartum, and to a larger set of acoustic descriptors. We show that none of these acoustic descriptors significantly differ between baby boys and girls, including fundamental frequency (perceived as pitch), perturbation and noise parameters, roughness, or cry duration. Critically, given that our study focused on short cries, we cannot completely rule out that acoustic features describing dynamic variation in these parameters over larger temporal scales may be related to sex. Nevertheless, our results suggest that idiosyncratic variations related to the individual identity of the baby are much more prominent markers of differences between babies than are possible sexed features.

**Cries change with age but retain an individual signature**. Our results highlight changes in infant cry acoustics with age. For instance, we found that babies' cries become more tonal from 0.5 to 3.5 months of age as acoustic entropy decreases and harmonicity increases with age for all three causes of crying. This result corroborates previous research showing a significant decrease in the noisy components of cries across the first 3 months of life[64]. This trend may be due to the maturation of the vocal folds[65] and/or to changes in the neurophysiology of crying as the infant's brain matures (see[5] for review). Another explanation for age-related changes may be that individual infants learn to cry in ways that most effectively arouse the attention of (and thus elicit aid from) their own parents or caregivers. Indeed, infants older than 2.5 months of age cry more predictably in response to external events that they have already encountered, compared to newborns whose cries appear more endogenously controlled[66]. Human infants rapidly increase their repertoire of complex vocal crying during the first 3 months of life[67] and subsequent months[68]. The possibility of learning is also supported by neuro-

psychological maturation with an increased understanding of the environment and thus, possibly, of nonverbal behaviors that most effectively attract caregivers[2,69].

In our study, we did not record as many crying babies at 3.5 months as at younger ages, as babies typically cry less frequently at older ages[1]. This may have weakened the classification power of the classifiers, and it cannot thus be excluded that at 3.5 months—and a fortiori at later developmental stages – babies' cries may become more informative about their causes. Manual corroboration of the specific acoustic results obtained here would be necessary in order to be confident that they align with underlying phonatory behaviors and physiologies. In future, it will also be important to study the developmental trajectory of crying over a broader age range and using more sensitive acoustic descriptives, including pitch contours and measures of temporal dynamics within bouts of crying.

Remarkably, we show that the drift observed in the acoustic characteristics of baby cries with age is individually predictable. Infant cries bear a very well-defined individual signature, allowing efficient identification of the baby by caregivers, across cry contexts. In two previous studies, our team showed that this individual cry signature and the possibility of its identification by caregivers are already present at birth[39] and at 2.5 months of age[40]. In particular, we previously showed that the ability to identify a baby by their cry is independent of the sex of the listener, as well as their parental status. Both women and men are able to identify a crying baby after a short exposure to its cries, but people with previous experience with babies perform better. Here, we show that this static information carried by the baby's cry predictably drifts over time. This predictability in the evolution of the individual acoustic signature may provide caregivers with a reliable adjustment of their ability to identify the baby throughout the baby's growth, especially in early ontogeny, as observed in other animals in which parental recognition of offspring is critical, such as seals and various bird species[70,71] (see[3] for a review). Human caregivers may seldom need to identify their baby through his or her cries alone. Nevertheless, they must constantly assess the baby's physiological state. The presence of a unique, predictably drifting vocal signature in a baby's cries may therefore provide caregivers with a stable reference point. Any deviation from this predictable trajectory of one or more of the acoustic features defining this signature could signal the onset of a possible problem deserving special attention. It can be hypothesized that the individual signature present in the baby cry has been selected in our ancestors for this purpose. Although they lived in small groups, where the probability of confusing one baby with another was probably small, they were able to assess the baby's condition by knowing the stable characteristics of their cries.

**Cries carry no specific contextual information**. Finally, our results support the hypothesis that the human baby cry is a graded signal of distress rather than a signal that codes for specific causes of crying[33]. Past studies testing this hypothesis focused on a small number of infants and a small number of acoustic parameters, recorded at one point in time and typically evoked using a trigger[72–76]. Here we employed an innovative methodology by recording 24 infants in real-life contexts over several months using a supervised automated method for cry segmentation, analysis, and classification. We retained only cries with sufficient signal quality and for which a parental report was given. This conservative approach led to the exclusion of some cries, for instance, excessive crying due to colic[77]. Thus, we focus on the three most commonly reported cry contexts: hunger, isolation, and discomfort, together representing 75% of the cries identified by parents. Only 0.2% of cries were labeled as pain cries, too few to examine their acoustic structure and perception. Without separating by age or baby identity, we found only very modest acoustic variation by cry cause. Notably, we did not find context-specific differences in any of the acoustic features considered, including fundamental frequency (pitch). Our study thus provides evidence that in the absence of other information (e.g., visual cues or contextual information such as time lapse between feeding events) and with the probable exception of pain, which we did not consider here, it is almost impossible to identify the cause of crying from individual cries alone.

Accordingly, our playback experiments provide evidence that naïve human listeners have major difficulties discriminating between babies' cries according to their ostensible causes, even following brief explicit training, and that the parental status of participants does not predict their ability to identify cry causes. It is noteworthy, however, that women performed slightly better than men did in our explicit training recognition task. One possible explanation could be that nonparent women in our sample had more experience with infants than did nonparent men, as women are often more involved from a young age in family tasks and cooperation in caring for infants[78]. Indeed, nonparent women in the explicit training experiment did indeed report more prior experience than did nonparent men. However, this imbalanced involvement of females and males in infant care is by no means a rule in the human species[78,79], and previous work has demonstrated that fathers, when they spend comparable time with their children, are as successful as mothers at recognizing their infant's cries[40]. It should also be noted that, although our psychoacoustic experiments did not reveal a significant ability of adult listeners to identify the cause of a cry, factors such as the age of the baby (the experiments were conducted with cries from babies aged 1.5 months only) or the duration of the listeners' training could impact this ability. It is therefore possible that highly trained listeners, with cries from older babies, could perform better.

**Limitations**. The present study has its limitations. As one cannot simply ask a preverbal baby why he or she is crying, we labeled the ostensible cause of each cry based on parental responses and behaviors that stopped the cry, and did not include rare and more extreme cry causes such as pain, wherein pain cries can be characterized by severe deviations from typical cry acoustic profiles[6,36] (see also[80] for an analysis of pathological sounds emitted by babies). By using a parental questionnaire, we obviously took the risk that parents might have a biased assessment of their babies' cries. However, the fact that there was a 75% correspondence between the parents' assessments and what actually stopped the cry suggests that this assessment was fairly valid. We also took the precaution of considering the action that stopped the cry rather than the parental assessment if the two did not match, which did not change our pattern of results.

In order to obtain high-quality cries for acoustic analysis and perception experiments, our results are based on brief individual cries extracted from longer cry sequences. While this ensured that our cry stimuli were devoid of noise and allowed for robust acoustic measurements, it is possible that focusing on long cry sequences may have produced different results. For example, we cannot exclude the possibilities that the number and duration of pauses in cry sequences lasting several seconds or minutes may be informative features or that information about cry cause may be encoded in the temporal, prosodic, or melodic structure of cry bouts, rather than in the spectrotemporal features of individual cries.

While our perception experiments corroborate earlier work showing that discriminating the cries of a given infant improves following brief familiarization[81], the ability to recognize an infant's cry among others depends mainly on the time spent with the child[39]. Cry recognition is thus likely to improve substantially with more extensive exposure and possibly also with exposure to longer cry bouts. We also highlight the fact that identification of cry context is typically multi-modal for parents, depending on visual, auditory, and olfactory cues as well as contextual cues (e.g., time of day or since feeding) that together may help parents to identify the cause of crying and resolve it in real-life contexts. Finally, as noted, the use of short cries may limit information about dynamic cry structure and temporal parameters, cues that parents may use to decide on caregiving interventions[11]. It would thus be worthwhile to conduct perception experiments using longer cry sequences to test the full extent to which human listeners can effectively decode contextual information from babies' cries. Moreover, by including cries from babies of various ages in listening experiments and varying listeners' time of exposure to those cries, researchers can test whether baby age and training intensity predict how well human listeners can decode cry cause by babies' cries.

## Conclusion

In conclusion, the present study suggests that the human infant did not evolve a universal discrete repertoire of cries to communicate the causes of its cries. Instead, each baby's cry encodes an individual signature that changes predictably with age and probably expresses general distress in a graded manner. While stable indexical signals to identity, sex, and age in the voice can be adaptive, for example, for parent-offspring recognition, one ultimate question remains: why haven't babies developed a universal discrete system for encoding contextual information in their cries? A first element of response is that a system using discrete signals to code for categories of information is not very flexible. Conversely, a strategy such as the one developed by human babies allows them to adapt their communication both to the causes of their cries (which can be diverse) and to the responses of their caregivers. This flexibility may contribute to the construction of individualized cry strategies such as the ones some of the babies in our sample may have developed. The condition for this system to be effective is obviously that the caregivers are familiar with the baby. This flexible coding seems rooted in our evolutionary tree as great ape infants seem to have developed a similar strategy[82,83]. The absence of discrete cry categories is thus reminiscent of what is observed in the vocal repertoires of other primate species. For example, bonobos show a graded structure of vocalizations with an individual vocal signature[84]. Unlike other so-called referential vocalizations, these graded acoustic signals do not allow for the designation of environmental elements or very precise contexts. This vocal flexibility is, therefore, characteristic of human infant

cries and reflects an evolutionary equilibrium of a dynamic communication system where both the sender and the receiver of the information must constantly adjust to the context as well as to their mutual interactions[85].

## Data availability

The raw data and source data supporting this article have been uploaded to the Open Science Framework (OSF) at https://osf.io/ru7na/ (https://doi.org/10.17605/OSF.IO/RU7NA).

## Code availability

All R codes supporting this article have been uploaded to the Open Science Framework (OSF) at https://osf.io/ru7na/ (https://doi.org/10.17605/OSF.IO/RU7NA).

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

## Acknowledgements
Funding was provided by the ANR (BABYCRY project n°ANR-19-CE28-0014-01, France), the Lyon IDEX Fellowship (D.R.), the Labex CeLyA (France), the University of Saint-Etienne (France), the Centre National de la Recherche Scientifique (CNRS, France), the Institut National de la Santé et de la Recherche Médicale (Inserm, France), the Swedish Research Council (Vetenskapsrådet; grant 2020-06352 to A.A.) and the Institut universitaire de France (F.L., N.M., and D.R.). The funders had no role in study design, data collection, analysis, decision to publish, or preparation of the manuscript. We thank the babies, their parents, and all the other adults who participated in this study.

## Author contributions
M.L.B., A.A., and K.P. have contributed equally to all aspects of this manuscript (data collection, sound analysis, statistical analysis, formatting of the database, writing of the manuscript). S.C. did the playback experiments. C.C. was involved in all aspects of the manuscript and was responsible for formatting the database. L.P. took part in the sound analysis. F.L. was involved in acquiring the recordings and analyzing the data. C.F. contributed to the playback experiments and data interpretation. H.P., N.M., and D.R. jointly supervised this work.

## Competing interests
The authors report no competing interests.
