## [Peer Review File · Communications Psychology]

7th Feb 23

Dear Professor Mathevon,

Thank you for your patience during the peer-review process. I am sorry for the delay in returning to you with a decision, which came about as a result of the holiday period and reviewer availability.

Your manuscript titled "What's in a cry? Stable and dynamic information in human baby cries" has now been seen by 3 reviewers, whose comments are appended below. You will see that they find your work of some potential interest. However, they have raised quite substantial concerns that must be addressed. In light of these comments, we cannot accept the manuscript for publication, but would be interested in considering a revised version that fully addresses these serious concerns.

We hope you will find the Reviewers' comments useful as you decide how to proceed. Should additional work allow you to address these criticisms, we would be happy to look at a substantially revised manuscript. If you choose to take up this option, please highlight all changes in the manuscript text file, and provide a detailed point-by-point reply to the reviewers.

Editorially, we consider three issues of utmost priority.

First, the referees highlight that as parents' ability to discern the cause of a cry is questionable, the model isn't trained/validated on a ground truth. If the information (regarding cry cause) provided by parents is unreliable, how can an algorithm trained with this unreliable data be expected to produce dependable outcomes? This needs to be highlighted throughout, mentioning the emerging caveats and limitations for interpretation, starting in the Abstract.

Second, the referees list a host of concerns regarding various analysis choices. All of these must be addressed through further analyses where possible, and a caveated discussion where necessary. The methodological approach should also be reported in more detail, especially with regard to the acoustic feature extraction and analysis. The underlying rationale should also be clarified.

Finally, the referees' report that the manuscript currently does not take the existing literature sufficiently into account (see recommendations by Reviewer 1 in particular). This must likewise be addressed, both in the Introduction and the Discussion.

If the revision process takes significantly longer than five months, we will be happy to reconsider your paper at a later date, provided it still presents a significant contribution to the literature at that stage.

We understand that due to the current global situation, the time required for revision may be longer than usual. We would appreciate it if you could keep us informed about an estimated timescale for resubmission, to facilitate our planning. Of course, if you are unable to estimate, we are happy to accommodate necessary extensions nevertheless.

We are committed to providing a fair and constructive peer-review process. Please do not hesitate

to contact us if you wish to discuss the revision in more detail.

Please use the following link to submit your revised manuscript, point-by-point response to the Reviewers' comments with a list of your changes to the manuscript text (which should be in a separate document to any cover letter) and any completed checklist:

[link redacted]

Please do not hesitate to contact me if you have any questions or would like to discuss the required revisions further. Thank you for the opportunity to review your work.

Best regards,

Jonna K. Vuoskoski

Jonna K. Vuoskoski, PhD
Editorial Board Member
Communications Psychology
orcid.org/0000-0003-0049-4373

EDITORIAL POLICIES AND FORMATTING

Editorial Policy: [Policy requirements](https://www.nature.com/documents/nr-editorial-policy-checklist.pdf) (Download the link to your computer as a PDF.)

Furthermore, please align your manuscript with our format requirements, which are summarized on the following checklist:

[Communications Psychology formatting checklist](https://www.nature.com/documents/commsj-psychol-style-formatting-checklist-article.pdf)

and also in our style and formatting guide [Communications Psychology formatting guide](https://www.nature.com/documents/commsj-psychol-style-formatting-guide-accept.pdf) .

*** TRANSPARENT PEER REVIEW:** Communications Psychology uses a transparent peer review system. This means that we publish the editorial decision letters including Reviewers' comments to the authors and the author rebuttal letters online as a supplementary peer review file. However, on author request, confidential information and data can be removed from the published reviewer

reports and rebuttal letters prior to publication. If your manuscript has been previously reviewed at another journal, those Reviewers' comments would not form part of the published peer review file.

* **CODE AVAILABILITY:** All Communications Psychology manuscripts must include a section titled "Code Availability" at the end of the methods section. In the event of publication, we require that the custom analysis code supporting your conclusions is made available in a publicly accessible repository; please choose a repository that provides a DOI for the code; the link to the repository and the DOI must be included in the Code Availability statement. Publication as Supplementary Information will not suffice. We ask you to prepare and upload code at this stage, to avoid delays later on in the process.

* **DATA AVAILABILITY:**

All Communications Psychology research manuscripts must include a section titled "Data Availability" at the end of the Methods section or main text (if no Methods). More information on this policy, is available at <http://www.nature.com/authors/policies/data/data-availability-statements-data-citations.pdf>.

At a minimum the Data availability statement must explain how the data can be obtained and whether there are any restrictions on data sharing. Communications Psychology strongly endorses open sharing of data. If you do make your data openly available, please include in the statement:

We recommend submitting the data to discipline-specific, community-recognized repositories, where possible and a list of recommended repositories is provided at <http://www.nature.com/sdata/policies/repositories>.

If a community resource is unavailable, data can be submitted to generalist repositories such as [figshare](https://figshare.com/) or [Dryad Digital Repository](http://datadryad.org/). Please provide a unique identifier for the data (for example a DOI or a permanent URL) in the data availability statement, if possible. If the repository does not provide identifiers, we encourage authors to supply the search terms that will return the data. For data that have been obtained from publicly available sources, please provide a URL and the specific data product name in the data availability statement. Data with a DOI should be further cited in the methods reference section.

REVIEWER EXPERTISE:

All Reviewers have expertise in acoustic analysis/modelling of human vocalisations; (perception of)

infant crying

Reviewer #1 (Remarks to the Author):

This study analysed 39,201 cries, recorded at home and in different contexts, from 24 babies (10 girls) aged between 15 days and 3.5 months. The authors describe having demonstrated that baby cries provide reliable information about age and identity, but “there is no evidence of cues to sex or cry cause”. Furthermore, they report that “neither machine learning algorithms nor trained adult listeners can reliably recognize causes of crying” and conclude that the “human baby cry is a graded acoustic signal that codes for need, without communicating a discrete cause or context.”

The study claims to address the issue of the information contained in the acoustic properties of the cries of babies aged from 15 days to 3.5 months. It is characterised by a complex design based on a comparatively large amount of data, which was analysed using very specific methods. This makes it different from many other studies on this topic and it thus has the potential to make an innovative contribution to the often-neglected field of pre-linguistic development in the context of language acquisition and communication. In this respect, I would very much support publication after the authors have better exploited the promise of their research by rewriting the manuscript. This is unsatisfactory in many places, because: the methods are not transparent; the interpretations of the results are not easy to follow; and the current state of the research in the field is not always taken into consideration. These concerns are explained in detail in the subsequent paragraphs.

Does the acoustic structure of baby cries differ by cause, either universally or within infants? (p.6, line134-135)

In the Introduction, the authors highlight that investigations of the issue of the causes of baby crying are insufficient, meaning that it is also unclear whether certain causes are encoded within acoustic properties at all.

This is a very important question, especially in light of the promised mobile apps for parents that falsely claim to employ AI to identify the causes of crying. The results of the present study, like earlier research, contradict the claims of such applications, and issues of child protection make it very important that relevant findings are published in an appropriate manner. In my opinion, this is the most important reason for investigating the causes of crying in greater detail. However, in relation to this manuscript:

- Earlier work is not cited, and nor are methods compared or discussed in the context of the authors’ own “novel” approaches to assessing baby cries. This is a serious problem methodologically and requires revision.
- All of the following have investigated the causes of crying using automatic classification methods, but are not referred to in the manuscript: Felipe et al., 2019; Sharma et al., 2019; Maghira et al., 2019; Franti et al., 2018; Osmani et al., 2017; Chang et al., 2016; Bano et al., 2015; and Yamamoto et al., 2013 a.o.

Several of these cited studies, as well as the authors' own perceptual experiments, suggest that parents are unable to identify causes of crying without additional contextual information. Consequently, why is this question raised repeatedly in the study? This is not explained in the Introduction.

In terms of the machine-learning approach applied in the study, there is a general methodological

problem that needs to be discussed: if the parental information (questionnaire etc.) is unreliable, then how is it anticipated that an algorithm trained with this unreliable data will be able to produce dependable outcomes?

Which acoustic parameters contribute to individual cry signatures? (p. 6, line132-133)

The authors argue that the lack of information on the causes of crying in baby sounds could be explained by selection pressures, which primarily promote the coding of individuality recognition. It is very welcome that explanations from evolutionary biology are employed to justify the selection of the entities used for the sound analysis. However, the argumentation has gaps and is not convincing:

→ Why would a hominin mother need cry characteristics to identify her baby?

→ The group size was small and social relationships were therefore easy to identify. In addition, the mothers probably carried their baby close to their body most of the time during the first three months of life. What evolutionary biological arguments are there for the temporary laying down of babies or allomothering in hominin ancestors, i.e., at a time when the cry characteristics are shaped?

It is true and acknowledged that, like all human beings, a baby's voice already has its own individual characteristics. These arise from neuro-physiological and anatomical features that do not prevent the encoding of "content" in vocal sounds. Indeed, humans would not have been able to develop a spoken language if this was not the case. Contrary to the authors' assumption, individual signatures in baby crying do not, therefore, contradict the fact that the cause of the crying is not encoded in the signal.

→ From my point of view, based on reliable information about identity, a comparison of the cry recognition attempts made by parents versus a machine algorithm might have been more appropriate than the efforts to recognise the causes of the crying.

However, despite the elaborate analysis in the study, there is little insight provided in the results about which parameters determine the identity of a baby.

→ The relevant question posed in the Introduction (p.6) is thus unanswered. Why?

The authors are mistaken in assuming that there are no universals in baby sounds at a given age. They exist simply because of how important these universals are for language acquisition and due to general maturation processes in the auditory-vocal system. Individual characteristics are of marginal value here. The conclusion on page 9, line 194-195, is neither correct nor supported by the data analysis. More restraint is thus required regarding the generalisation of the findings.

Are infant cries sexually dimorphic? (p.6, line 132)

The approach to identifying biological sex based on acoustic properties is not reasoned substantively in the manuscript. Why should infant cries be sexually dimorphic? Indeed, clear physical characteristics mean that a selection advantage does not seem to exist here.

All aspects of "sex" discussed in the manuscript require consideration of recent findings on the influence of hormones on voice characteristics in babies. (s. review in *Horm Behav* 2018 Aug; 104:206-215. doi: 10.1016/j.yhbeh.2018.03.008).

→

Methods

Extraction of cry syllables

→ Could the authors report the values for selecting “excellent or acceptable signal-to-noise ratios”? Was this decided per session, per sequence or per syllable level and how?

→ The segment duration is not reported in Table S2, contrary to what is stated on page 23, line 499.

→ How were “cry syllables” defined? Does a cry syllable refer to the same thing as a cry sequence? How is “session” defined? A clear explanation is required here, as the authors base their analyses on “cry syllables” (page 23).

→ How were longer sequences segmented manually? What criteria were used to decide where to place the segmentation cursors in PRAAT to extract the "syllables" from longer sequences? This is important because cry bouts are very variable in their duration, ranging from 1- 20 seconds and sometimes even longer.

→ The description on page 23/24, lines 500-513, is confusing and requires rewriting. It appears as if, in a first step, suitable cry sequences ($n=676$) of variable durations ($49\pm 73s$) with an adequate signal-to-noise-ratio were selected manually. This does not seem to refer to cry syllables, as 676 “were automatically segmented into 78094 individual cry syllables” (line 506-507)? How were cries (Table S2) therefore defined here?

→ In general, I am skeptical about the entire segmentation process. I have been working in this field for more than 30 years and, based on my (and other authors’) experience, the correct segmentation of cry utterances is one of the most complicated procedures in the field.

→ A complex, partly unclear, segmentation procedure is described, but it is not obvious what was actually left at the end and what was evaluated. Why was there not simply segmentation using PRAAT? This would not only have provided the greatest certainty about what the segments would ultimately look like, but it would also be verifiable. Why are there so many methods mentioned just for the segmentation? The approach requires a detailed explanation. Sound examples (supplements) might also aid understanding.

→ What does a minimum of 20% voiced mean? This seems to be very low harmonicity. Baby cries typically have unvoiced regions, both subharmonic and shifts. Were unvoiced elements relevant in the "information in the cry" sought by the authors? Accordingly, exactly what was done here and why needs to be explained.

→ The criterion "median pitch > 150 Hz" is obscure. Baby sounds have a median pitch of about 400 Hz. What has been evaluated here? Grunts with a lower frequency were excluded. In my opinion, the criterion must have led to analysis errors, despite the claim of "examining the ratio of false syllables". In my view, this is a serious methodological problem.

→ Does it really make sense to evaluate "jitter" (and "shimmer") when considering the recording conditions? How reliable are the values?

Acoustic analysis

→ Again, how are the cry syllables defined, each of which is said to have been analysed with 10 key acoustic variables? What is the difference between a cry syllable and a cry (Table S2)?

→ The selection of the acoustic variables is not explained. The comment "known to be highly informative" is insufficient.

The selection and significance of the acoustic measurements must be put in the context of the study's issue of concern and be supplemented with theoretical considerations. Hardly anything is said about this. The coherence of the content of the individual parts of the manuscript suffers as a consequence.

RESULTS

→ To better assess the results, the outcome data of the acoustic measurements would be very helpful. However, I suspect that these are not available.

→ Depending on age, acoustic entropy and harmonicity were found to be most important. What is meant by acoustic entropy? Could you provide typical examples?

→ Which parameters determined identity in the sample?

→ In the introduction, page 3, lines 74-76, it is stated: "For example, human infant cries are often characterized by nonlinear phenomena that arise from aperiodic vibration of the vocal folds and that contribute to the perceived roughness of vocalizations (Fitch et al., 2002)." The paper by Mende et al. (1990), who first described those phenomena, requires citation here: "Bifurcations and chaos in newborn infant cries." (Physics Letters A, Volume 145, Issues 8–9, 30 April 1990, Pages 418-424).

Overall, to demonstrate that this is a solidly designed, theoretically well-grounded study, thorough revision is strongly warranted.

Reviewer #2 (Remarks to the Author):

I'm pleased to have read this submission and have no major concerns regarding the validity of its conclusions, being primarily that 0-4 mo old infant cries signal their identities, and even their distinctive ways of aging, but not their proximal causes (e.g., hunger, isolation, physical discomfort). These findings depend on algorithm-based explorations of a large dataset of cry 'syllables,' relying on an acoustic synthesis program for acoustic parameter extraction and on sophisticated and innovative Random Forest-based analyses of these parameters. The cause observations were also corroborated perceptually, finding that virtually no gender/experience category of listener reliably detected these causes. The author group had previously found that similar groups of listeners were able to identify babies (though I recommend that it's findings be highlighted more so in this report's discussion, as so closely complementary to the current findings).

In lauding these conclusions above, however, I also allude to some concerns with the reporting, and underlying procedures. These concerns arise from my professional perspective as an acoustician closely concerned with human vocal development (and, to further qualify my remarks, my focus has been on non-cry speech-like vocal development in the first year of life). I respect that the author group and the literature cited here is primarily ethological, but to reach and align with human speech development research I strongly recommend that they reconsider use of the term 'syllable' in reference to their cry units. Similarly respectful of the authors' intended contributions, a limitation towards not just delivering clear information to developmentalists in terms that may link to underlying behaviors and physiologies, it must be acknowledged that the reliance on an automatic algorithm (with no apparent validation against any ground truth measures or 'gold standard' human coding), does not permit a clear definition of the units measured here.

That being said, 'continuous non-background sound' probably associates with exhalatory breath groups (though possibly including rhythmic inhalatory/exhalatory cycles, but probably excluding many extended glottal holds, i.e., breaking the surrounding phonation into distinct units). Regardless of whether these speculations are grounded (though I'm confident that anyone listening to the units could gain a good impression), these are not syllables. The onset of marginal and canonical syllables, specifically involving consonant-vowel or vowel-consonant transitions, all much shorter than the range of durations reported here, is arguably the single most important milestone in human vocal development. In the present case, because I found no discussion of any theoretical or ontological basis for the cry unitizing, I sense that this is actually a minor issue of nomenclature. Suggested short substitutes for breath group might be phrase, utterance, vocalization (or even simply 'cry' and 'cries,' with no further ascription).

Returning to the broad application of algorithms, first for parameter extraction and secondly for data analyses, I need to acknowledge that I have no direct expertise with the analysis approaches; The outcomes are adequately presented and reasonably interpretable in statistical and graphic forms.

Regarding the acoustic analyses, however, I felt all could have been more carefully couched in terms of the limitations imposed by algorithmic approaches and their own control parameters. It's a bit regrettable, that though they did use the well-documented and authoritative phonetic analysis program Praat for some parameter extractions, the authors relied instead on a synthesis package for most. Yet with any program, constraints on an f_0 extraction algorithm (PDA), will have a significant impact on the representativeness and distribution characteristics of the dataset. Of greatest relevance here, the preponderance of non-modal vibratory regimes found in infant phonation literally begs questions of f_0 definition. Considering that the range of possible glottal periodicities is not just on the order of multiple kHz but also down to 10's of Hz, f_0 defined as such requires supervision of PDAs. That was clearly not the case here, perceived pitch is a completely different level of interpretation, yet considering chaotic, subharmonic, and creaky voices the authors recognize as characteristic of crying (especially early crying), f_0 data obtained in the manner here can't be interpreted as reflecting actual behavioral central tendencies or ranges. In fact, without proper parameter adjustments, the pressed nature of low frequency pulses falls below floor values and is then misread at quite high frequencies due to the high attack rate of these pulses, yielding measurement tendencies that move in exactly the wrong directions.

This may even have something to do with the counterintuitive finding that f_0 increased for most of

these infants with age is counterintuitive, especially in light of relatively impressionist remarks in the manuscript that older cries were less ‘shrill.’ My simplest request is that the authors include statements regarding limits of the approach that affect interpretation. Yet ideally, the identification of some ground truth criterion may help future efforts, calibrating against effects due to algorithmic defaults, and stepping more closely to clinical desirable understanding of the underlying physiology, cf. (Hirschberg et al., 2008).

Finally, it might have been helpful to consider more carefully what speech-like utterances were captured in this dataset or how systematically they were truly eliminated—there are certainly many more such sounds than ‘grunts,’ which some take to refer to merely mechanical or ‘vegetative’ origins; there’s much related literature, too much to include, except I wish to point out just one which is so similarly in an animal ethology tradition that I think it might be considered here—I am not an author or member of this group nor involved with the prior citation: (Scheiner et al., 2002).

Hirschberg, J., Szende, T., Koltai, P., & Illenyi, A. (2008). *Pediatric Airway: Cry, Stridor and Cough*. Plural Publishing.

Scheiner, E., Hammerschmidt, K., Jurgens, U., & Zwirner, P. (2002). Acoustic analyses of developmental changes and emotional expression in the preverbal vocalizations of infants. *Journal of Voice*, 16(4), 509. <http://www.sciencedirect.com/science/article/B7585-487TGP3-27/2/3f7a034f24d6c6feded2c86cda9e96c7>

Reviewer #3 (Remarks to the Author):

The research presented in the manuscript titled “What’s in a cry? Stable and dynamic information in human baby cries” aims to identify if a set of acoustic features measured in infant cry bouts can be used to accurately classify babies’ identity, age or sex, as well as the cause of the cry.

To investigate this, the authors combined modeling with psychoacoustic experiments in two samples of adults, measuring the acoustic characteristics of cries in a longitudinal sample, with babies being recorded on 48 hours measurements from 2 weeks to 4 months. The cause of the cry was also recorded.

Findings show that the models could classify above chance the identity and age of the babies starting from their cry acoustic features, but they were not able to do so for babies’ sex and for the cry cause.

Similarly, two samples of adults, both implicitly and explicitly trained could not classify the cry causes above chance. It is also important to notice that the ability to classify cries was significantly moderated by adults’ own experience with infants.

Two more interesting findings showed that (i) cries features are significantly different between individuals while stable within one individual across time and that (ii) there were some universal changes to cry characteristics with age (i.e. cry becomes more tonal).

The approach used here is innovative and the research question represents an important advance in understanding and has important practical applications. This research project helps to objectively test an impactful bit of common knowledge which, if understood better, could improve the advice given to parents as well as parents’ own experience and well-being in the first months of their baby’s life.

The statistical analysis is appropriate and the level of detail provided in the methods will allow other researchers to reproduce the work.

I recommend the manuscript for publication pending a few minor revisions (see below).

1. Across the whole manuscript the authors claim that their findings show that it is not possible to classify a cry's cause based on its acoustic features alone. However, they themselves highlight that individuals' cries vary significantly and that experience with infants in general as well as with a particular infant is important in determining the cause of a cry. Moreover, the cry itself changes with age and it becomes, as written in the manuscript, more dependent on context while the baby grows. I am aware that models including the individual and the age as factors, as well as their interaction, were also run with no real improvement in the classification accuracy. However, not all the individual babies gave samples for all timepoints and this might have undermined the possibility to classify stable features in individual babies across time. Moreover, the latest time point, which could have been the one with the most structured cries, had fewer babies and this could have undermined the classification power and the most reliable – because of the cry stronger relation to context – timepoint.

In the psychoacoustic experiments, adult participants were given some experience with a particular baby before being tested on the same baby. However, it might be necessary to have more experience with one particular baby or, alternatively, the failed classification might again be linked to age. Adult participants listened to cries of babies at 1.5 months. But again, cry might take longer to become contextualised. Therefore, results could change when considering older babies only or by testing their own parents.

Lastly, no dynamic acoustic features (i.e. envelope, number and length of pauses, etc) were included in the analysis and these might carry important information about the type of cry or the context. The findings from the paper are solid and the authors do address some of these considerations in the limitation section. However, the language used overall in the manuscript makes the reader think that the results about the cry causes are more definite and that in general the cause of a cry cannot be determined from the cry itself ever. I would appreciate this to be smoothed out by leaving more room for doubt or for different factors.

2. Discussion, line 379. The authors "predict" what an abrupt deviation from the usual acoustic feature might signal. I am not clear why a prediction is being made here in the discussion.

3. In the parent questionnaire on cry cause, parents were asked both their opinion on the cry cause and which action made the cry stop. In 75% of cases the two overlapped. Where they did not agree, the action that made the cry stop was considered the cry cause for the purpose of the experiment. I am wondering what would happen to the model classification accuracy of cry cause if the parental opinion was used instead. I do not really expect it to change much, but it would be a nice control to have.

4. The training and testing session in the psychoacoustic experiments featured cries from the same baby. Was it the same baby for all participants?

5. Which version of the analysis software (i.e. Praat, R) was used? It is important information for other researchers who would like to reproduce the work.

Response to Referees

Reviewer #1 (Remarks to the Author):

R1_1 This study analysed 39,201 cries, recorded at home and in different contexts, from 24 babies (10 girls) aged between 15 days and 3.5 months. The authors describe having demonstrated that baby cries provide reliable information about age and identity, but “there is no evidence of cues to sex or cry cause”. Furthermore, they report that “neither machine learning algorithms nor trained adult listeners can reliably recognize causes of crying” and conclude that the “human baby cry is a graded acoustic signal that codes for need, without communicating a discrete cause or context.”

The study claims to address the issue of the information contained in the acoustic properties of the cries of babies aged from 15 days to 3.5 months. It is characterised by a complex design based on a comparatively large amount of data, which was analysed using very specific methods. This makes it different from many other studies on this topic and it thus has the potential to make an innovative contribution to the often-neglected field of pre-linguistic development in the context of language acquisition and communication. In this respect, I would very much support publication after the authors have better exploited the promise of their research by rewriting the manuscript. This is unsatisfactory in many places, because: the methods are not transparent; the interpretations of the results are not easy to follow; and the current state of the research in the field is not always taken into consideration. These concerns are explained in detail in the subsequent paragraphs.

We are very grateful to the referee for their thorough review of our work. It is indeed a rather complex study based on a large amount of data and extensive analysis methods. We agree that many points needed to be explained in much greater detail. This is what we have done in this revised version. However, we remain at the referee's disposal for further clarifications or explanations.

R1_2 Does the acoustic structure of baby cries differ by cause, either universally or within infants? (p.6, line134-135)

In the Introduction, the authors highlight that investigations of the issue of the causes of baby crying are insufficient, meaning that it is also unclear whether certain causes are encoded within acoustic properties at all.

This is a very important question, especially in light of the promised mobile apps for parents that falsely claim to employ AI to identify the causes of crying. The results of the present study, like earlier research, contradict the claims of such applications, and issues of child protection make it very important that relevant findings are published in an appropriate manner. In my opinion, this is the most important reason for investigating the causes of crying in greater detail. However, in relation to this manuscript:

- Earlier work is not cited, and nor are methods compared or discussed in the context of the authors' own “novel” approaches to assessing baby cries. This is a serious problem methodologically and requires revision.

We agree with the referee that knowing whether cries carry information about their cause is a very important goal. Our study should thus contribute to weakening the commercial arguments put forward by vendors of AI applications with false claims of decoding cries.

Following the referee's comment, we have added a substantial sub-section in the introduction (lines 72-135) discussing previous work (including most papers noted by the referee, see R1_3) and placing our study in context. We insist in particular on the poverty of the available recording banks, which is a fact underlined in the whole literature. Although many powerful acoustic analysis and machine learning methods have recently been developed, most of them could not previously be appropriately validated due to the absence of archives of healthy babies' cries, of the same age, and with well-documented identity of the babies and causes of their cries. An essential element - and a new one compared to previous studies - is therefore that we worked with a substantial databank of cries that we recorded longitudinally (babies were recorded several times during the first 3 months) and in real-life contexts.

R1_3 All of the following have investigated the causes of crying using automatic classification methods, but are not referred to in the manuscript: Felipe et al., 2019; Sharma et al., 2019; Maghfira et al., 2019; Franti et al., 2018; Osmani et al., 2017; Chang et al., 2016; Bano et al., 2015; and Yamamoto et al., 2013 a.o.

We now cite all these references (except Yamamoto et al. 2013, which we find out of scope).

R1_4 Several of these cited studies, as well as the authors' own perceptual experiments, suggest that parents are unable to identify causes of crying without additional contextual information. Consequently, why is this question raised repeatedly in the study? This is not explained in the Introduction.

In fact, we think this question is still open. On one hand, there are not that many studies that have tested whether adults can identify the cause of a cry, and most of these studies have focused on highly distinct cry contexts (e.g., pain versus mild distress). On the other hand, previous studies have not tested the hypothesis that each baby can develop their own strategy for encoding information, nor has any previous study tested whether decoding cries can be learned. Finally, the idea that parents can identify the cause of their baby's cries is widespread in the general public. It is for these reasons that we conducted a longitudinal study, with babies recorded several times during the first 3 months of life. This originality allows us to test the possibility that each baby may have its own coding strategy for encoding the cause of their crying. Another important aspect is that we focused on the most common, and the most ecologically relevant, causes of crying, omitting extreme pain. In this revised manuscript, we explain this further in the Introduction (lines 60-81):

“Traditionally, infant cries were thought to represent acoustically distinct cry types, each associated with a discrete cause such as birth cries, pain cries, hunger cries, pleasure cries, startle cries, and attention cries (Müller et al., 1974; Wasz-Höckert et al., 1964). Early studies suggested that mothers and trained nurses could identify such cry types without additional contextual cues (Wasz-Höckert et al., 1964; Wiesenfeld et al., 1981). However, recent research suggests that identifying discrete information related to the cause of a cry is not so straightforward. For example, while parents can discriminate between highly distinct cry contexts, such as pain versus mild discomfort (Koutseff et al., 2018), discrimination is substantially degraded for cries that share a similar level of distress (Gustafson & Harris, 1990). Moreover, it has been shown experimentally that an adult listener's ability to classify a cry as

communicating pain or discomfort is highly dependent on their prior experience with infants (Corvin et al., 2022).

Although these previous studies suggest that it is difficult to identify the cause of a cry by ear, it is commonly believed, especially by the general public, that parents can discern why their baby is crying just by listening. Books on parenting echo this belief, while countless websites offer recipes to decipher babies' cries. Some non-academic sources even suggest that babies' cries are a "language" made up of phonemes whose meaning can be learned, and mobile applications proposing to decode babies' cries are becoming increasingly popular, despite a lack of fundamental scientific evidence to support their veracity. Moreover, researchers have yet to test the hypothesis that each baby may develop his or her own coding strategy for the cause of crying, as we test here. The question of whether baby cries encode information about their cause has therefore not yet been effectively answered.”

R1_5 In terms of the machine-learning approach applied in the study, there is a general methodological problem that needs to be discussed: if the parental information (questionnaire etc.) is unreliable, then how is it anticipated that an algorithm trained with this unreliable data will be able to produce dependable outcomes?

Identifying the real cause of a cry is not easy of course, as the baby is not able to tell us directly. We therefore used a parental questionnaire. As noted above, in this questionnaire, parents were asked to suggest a cause for each cry made by their baby, but were also asked to indicate what action had stopped the cry, and in 75% of the cases the action that stopped the crying matched the parental assessment of the cry cause (for example, parents indicated "hunger," and feeding their child did indeed stop the cry). When the parent's assumed cause did not match the action that stopped the cry, we coded the cause of the cry based on that action and not the parental assessment.

This method seems to us to be the most likely to identify the cause of the cry in a reliable, bottom-up manner. We also took the precaution of running our models on both the parental indications and what actually stopped the cry. Our results show that the difference in classification between the two approaches is negligible (see R3_6).

We have now clarified this in the discussion section (lines 527-532) and in the methods section lines 608-619, as follows:

“During each 48-hour recording period, parents completed a form indicating, for each crying sequence produced by their baby, the onset time of the cry, the potential cause identified for the cry, the action(s) taken to stop the cry, and the action that was ultimately effective. Parents chose from the following causes: hunger, isolation, physical discomfort (such as fever, cold temperature, full diaper), pain, and unknown. There was also an option for parents to indicate a cry cause that was not listed via an open response comment box. In 75% of the cases, the action that stopped the crying matched the parental assessment of the cry cause (for example, parents indicated "hunger," and feeding their child did indeed stop the cry). When the parent's assumed cause did not match the action that stopped the cry, we coded the cause of the cry based on that action and not on the parental assessment. We made this choice to increase the reliability and objectivity of labeling cry causes.”

See also new text in Results, lines 322-324:

“Using the cause of crying indicated by the parents in the questionnaire, instead of the action that stopped crying, produced nearly identical results with a classification accuracy of 35%, 95% CI [33, 36]. This is not meaningfully different from 36% [33, 38] based on the cause that stopped crying.”

R1_6 Which acoustic parameters contribute to individual cry signatures? (p. 6, line 132-133)

All ten acoustic characteristics noticeably contributed to individual recognition of babies by Random Forest models, but median pitch was the top predictor, as we now clarify in the Results (lines 275-277). This is in line with the finding that babies' voices have stable pitch levels relative to their same-aged peers (Fig. 2C).

R1_7 The authors argue that the lack of information on the causes of crying in baby sounds could be explained by selection pressures, which primarily promote the coding of individuality recognition.

It is very welcome that explanations from evolutionary biology are employed to justify the selection of the entities used for the sound analysis. However, the argumentation has gaps and is not convincing:

- Why would a hominin mother need cry characteristics to identify her baby?
- The group size was small and social relationships were therefore easy to identify. In addition, the mothers probably carried their baby close to their body most of the time during the first three months of life. What evolutionary biological arguments are there for the temporary laying down of babies or allomothering in hominin ancestors, i.e., at a time when the cry characteristics are shaped?

We agree with the referee that identifying one's baby among others on the basis of its cries was probably not necessary in small groups of ancestral hominins. On the other hand, knowing the individual characteristics of one's own baby's cries was and still is useful in identifying possible problems the baby might be experiencing. The individual vocal signature is a stable marker. Any change in this signature can be a marker of illness or accident. The hypothesis that the individual signature present in the cries results from, and is conserved by, natural selection thus makes sense. We have clarified this view in the discussion (lines 475-484):

“Human caregivers may seldom need to identify their baby through his or her cries alone. Nevertheless, they must constantly assess the baby's physiological state. The presence of a unique, predictably drifting vocal signature in a baby's cries may therefore provide caregivers with a stable reference point. Any deviation from this predictable trajectory of one or more of the acoustic features defining this signature could signal the onset of a possible problem deserving special attention. It can be hypothesized that the individual signature present in the baby cry has been selected in our ancestors for this purpose. Although they lived in small groups, where the probability of confusing one baby with another was probably small, they were able to assess the baby's condition by knowing the stable characteristics of their cries.”

R1_8 It is true and acknowledged that, like all human beings, a baby's voice already has its own individual characteristics. These arise from neuro-physiological and anatomical features

that do not prevent the encoding of "content" in vocal sounds. Indeed, humans would not have been able to develop a spoken language if this was not the case. Contrary to the authors' assumption, individual signatures in baby crying do not, therefore, contradict the fact that the cause of the crying is not encoded in the signal.

To clarify, we are not saying that the presence of an individual signature in the cry contradicts the fact that the cause of the cry is not encoded in the signal. On the contrary, we emphasize that cries are characterized by both stable and variable acoustic features, and noted several times throughout the manuscript. The aim of our study is precisely to understand how these features develop throughout the first months of life. This is specified in lines 188-192:

“In this longitudinal study, we test for acoustic variability and stability in human baby cries, both across infants and across the first four months of each infant's life. We test the predictions that babies have individual cry signatures, and that these individual cry signatures do not differ between the sexes and remain stable with age. We also test the prediction that there are consistent acoustic differences between baby cries caused by different events.”

In our study we also test the possibility that each infant, with its individualized cry profile, might also encode for context or cry cause in a consistent but individualized manner, again supporting the notion that individual cry signatures may still give rise to context-specific cries at the level of the individual. We did not, however, find evidence to support this.

R1_9 From my point of view, based on reliable information about identity, a comparison of the cry recognition attempts made by parents versus a machine algorithm might have been more appropriate than the efforts to recognise the causes of the crying.

We have already tested whether adult listeners including parents can recognize a baby by its cry in two previous studies:

-Gustafsson et al. 2013. Fathers are just as good as mothers at recognizing the cries of their baby. *Nature Communications*, 4, 1698.

-Bouchet et al. 2020. Baby cry recognition is independent of motherhood but improved by experience and exposure. *Proceedings of the Royal Society B*, 287, 20192499.

In those papers, we have shown that such cry recognition is robust, and that adult listeners quickly learn to identify a baby by its cries.

We specify this in lines 154-158:

“Indeed, cries have long been known to differ acoustically from one baby to another (Gustafson et al., 1994, 2017), and we have experimental evidence of the ability of parents and naive listeners to recognize specific infant calls very reliably (Bouchet et al., 2020; Gustafsson et al., 2013). This is why it is critical to control for baby identity when testing whether baby cries communicate their cause.”

R1_10 However, despite the elaborate analysis in the study, there is little insight provided in the results about which parameters determine the identity of a baby.

- The relevant question posed in the Introduction (p.6) is thus unanswered. Why?

We agree with the referee and now mention the acoustic parameters that encode the identity of a baby (lines 275-277).

Please see our response to R1_6.

R1_11 The authors are mistaken in assuming that there are no universals in baby sounds at a given age. They exist simply because of how important these universals are for language acquisition and due to general maturation processes in the auditory-vocal system. Individual characteristics are of marginal value here. The conclusion on page 9, line 194-195, is neither correct nor supported by the data analysis. More restraint is thus required regarding the generalisation of the findings.

We agree with the referee's comment that human infant cries share universal characteristics, which does not preclude them from having idiosyncratic features that define an individual signature. We have rewritten this conclusion as follows (lines 277-279):

“These results therefore support the hypothesis that babies' cries carry idiosyncratic acoustic characteristics that define an individual signature, unique to each baby (Gustafson et al., 1994, 2017).”

R1_12 Are infant cries sexually dimorphic? (p.6, line 132). The approach to identifying biological sex based on acoustic properties is not reasoned substantively in the manuscript. Why should infant cries be sexually dimorphic? Indeed, clear physical characteristics mean that a selection advantage does not seem to exist here. All aspects of "sex" discussed in the manuscript require consideration of recent findings on the influence of hormones on voice characteristics in babies. (s. review in *Horm Behav* 2018 Aug; 104:206-215. doi: 10.1016/j.yhbeh.2018.03.008).

We tested whether infant cries carry a sex signature because our sample of babies had a mix of girls and boys. However, as noted in our introduction, we did not predict an acoustic signature of sex, nor did we find one. We now justify more precisely why this is very likely the case, relying in particular on the review proposed by the referee (lines 170-176):

“Because sexual dimorphism of the vocal anatomy does not emerge until puberty in humans (Titze, 1989), we do not expect sex differences to be present in baby cry acoustics. Indeed, in a previous study, we found no difference between the pitch of female and male baby cries (Reby et al., 2016). Wermke et al. (2018) also point out that sex does not directly predict variance in cries, however peripheral estradiol concentrations (a baby-specific characteristic) do predict infant vocal performance. In the present study, we did not investigate hormone levels, but we did test for effects of sex and age.”

R1_13 Methods

Our general response to the referee's comments about Methods:

We thank the reviewer for several constructive comments regarding the editing and selection of cry sequences and syllables. Following these comments (see below), we have revised the manuscript to improve clarity, transparency and reproducibility of this process. Specifically,

we now describe each of the six steps in the cry editing and selection process, employed to ensure that our final cry stimuli were devoid of noise. We also provide operational definitions of our terms, the use of which we have now standardized throughout. These descriptions are aided by a new illustration of the step-by-step process (Figure 4).

Please also note that the R script used to segment cry sequences into individual cries in *soundgen* (along with all other R code, scripts, datasets and output) are openly available on the Open Science Framework (<https://osf.io/ru7na/>).

Here we briefly describe in bullet points the multi-step process, and below we respond in more detail and in line to each of the reviewer's specific queries and concerns. The cry selection and editing process is now also described in more detail in the Methods section ('Extraction of cries' lines 621-680) and in the new Figure 4.

- In step 1, we manually segmented each 48-hour audio recording (taken from each baby at each age), into cry sequences. Each cry sequence corresponded to a single questionnaire entry (regarding cry cause) completed by the parents, and thus to a single crying bout.
- In step 2, these cry sequences were evaluated manually (visually from spectrograms and acoustically), and those with a high level of noise (i.e., wherein most of the sequence was characterized by overlapping background noise) were discarded, while those with little to no noise ("excellent or acceptable signals") were retained.
- In step 3, these retained cry sequences were cleaned by manually cutting out background noise. This resulted in clean, spliced cry sequences.
- In step 4, these sequences were automatically segmented into individual vocalizations ('cries', previously labelled 'cry syllables'). This was done using the customized *segment* function in *soundgen*, as described in the text (p. 31) and in more detail in response to the reviewer's comments below.
- In step 5, to retain only those vocalizations that corresponded to cries while omitting any other non-cry vocalisations (e.g., grunts, coughs) for acoustic analysis, vocalizations were selected on the basis of four criteria: (1) minimum 20% voiced; (2) median pitch > 150 Hz; (3) duration > 250 ms; and (4) wiener entropy < 0.6 s. As explained in the manuscript, inclusion criteria were extensively piloted and these specific criteria yielded the lowest possible rate of false positives (non-cry vocalisations) in the selected sample (2%).
- Finally, as a 6th and final step of the cry selection process, we focused our analyses on individual cries associated with the most common causes: hunger, isolation, or discomfort.

R1_14 Extraction of cry syllables

- Could the authors report the values for selecting "excellent or acceptable signal-to-noise ratios"? Was this decided per session, per sequence or per syllable level and how?

As described in R1_13, noise was evaluated in two steps. As the process was manual in both steps, noise was identified as any overlapping background sound that occluded the cry of the infant, and which would thus interfere with acoustic analysis. Noise corresponded in most cases to parental voices or distant sounds from televisions, music, etc. First (step 2) noise was identified visually and acoustically from spectrograms for each of 3308 cry sequences in order to categorize each sequence as excellent, acceptable, or unacceptable (see R1_13). At this

stage we retained only those cry sequences coded as acceptable or excellent. Then, in step 3, we manually removed any remaining noisy segments from the retained cry sequences.

R1_15 The segment duration is not reported in Table S2, contrary to what is stated on page 23, line 499.

We thank the reviewer for bringing this to our attention. Indeed, the average duration of the cry sequences is given in the text, whereas Table S2 provides summary data on cry ‘syllables’ (now referred to as ‘cries’). We have thus removed the reference to Table S2 from this line in the text. We have also revised Table S2 to standardize the terminology.

R1_16 How were “cry syllables” defined? Does a cry syllable refer to the same thing as a cry sequence? How is “session” defined? A clear explanation is required here, as the authors base their analyses on “cry syllables” (page 23).

We have now added operational definitions for these terms within the manuscript, and as noted above, we have also created a figure to aid in the interpretation of the cry editing and selection process (Figure 4). Specifically, we define the terms as follows:

Recording session: The 48-hour periods during which continuous audio recordings were taken within each infant’s home environment, at up to 4 time points/ages (0.5, 1.5, 2.5 and 3.5 months of age).

Cry sequence: Cry sequences were extracted from the full 48-hr recording sessions for each infant at each age. Each cry sequence corresponded to a single questionnaire entry (regarding cry cause) completed by the parents, and thus to a single crying bout.

Cry syllables/cries: To avoid confusion with common linguistic interpretations of ‘syllables’ (as suggested by reviewer 2), we have changed ‘cry syllables’ to ‘cries’ throughout the manuscript. Cries were extracted as continuous, at least partly voiced episodes of vocalizing that were different from steady background noise and satisfied additional criteria (to exclude non-cry vocalizations), as described in the methods (see R_13 and Figure 4).

R1_17 How were longer sequences segmented manually? What criteria were used to decide where to place the segmentation cursors in PRAAT to extract the "syllables" from longer sequences? This is important because cry bouts are very variable in their duration, ranging from 1- 20 seconds and sometimes even longer.

The process by which cry sequences were segmented into cries using the *segment* function in *soundgen* is described in the manuscript (lines 635-640; Fig. 4). Extensive details on the segmentation algorithm are available in the documentation of the *soundgen* function *segment*, and the R code for segmentation is provided in the supplements (scripts.zip, file prep.R).

Briefly, cries were extracted from crying sequences using the *segment* function in *soundgen* (Anikin, 2019). This algorithm transforms a long recording into a mel-spectrogram, estimates noise spectrum by means of locating a relatively quiet and stable acoustic component, calculates signal contour as the inverse of cosine similarity between the spectrum of each frame and the estimated spectrum of noise weighted by amplitude, and then detects continuous syllables as "islands" of this derived signal contour on a particular time scale. In this case, we looked for segments of minimum 50 ms in duration, separated from other cries by at least 100 ms.

To avoid excessive technical explanations, we have simplified the description of the automatic segmentation procedure in the text, instead providing explicit references to *soundgen* documentation and the segmentation script in the supplements to allow for replication.

R1_18 The description on page 23/24, lines 500-513, is confusing and requires rewriting. It appears as if, in a first step, suitable cry sequences ($n=676$) of variable durations ($49\pm 73s$) with an adequate signal-to-noise-ratio were selected manually. This does not seem to refer to cry syllables, as 676 “were automatically segmented into 78094 individual cry syllables” (line 506-507)? How were cries (Table S2) therefore defined here?

We hope that this is now clear given our above explanations. As suggested, we have now revised several sections of the manuscript text, including the text that was previously found on pages 23-24, to clarify and provide further details regarding the cry selection and segmentation process (see also Figure 4). We have therefore revised the whole text of the section “Extraction of cries” (now lines 621-680). For clarity we have also revised the Table column headings in Table S2 to specify the cry unit, changing “cry syllables” to “cries”. Please see R1_13 for more details.

R1_19 In general, I am skeptical about the entire segmentation process. I have been working in this field for more than 30 years and, based on my (and other authors’) experience, the correct segmentation of cry utterances is one of the most complicated procedures in the field. - A complex, partly unclear, segmentation procedure is described, but it is not obvious what was actually left at the end and what was evaluated. Why was there not simply segmentation using PRAAT? This would not only have provided the greatest certainty about what the segments would ultimately look like, but it would also be verifiable. Why are there so many methods mentioned just for the segmentation? The approach requires a detailed explanation. Sound examples (supplements) might also aid understanding.

We fully agree with the reviewer that segmenting cry utterances is no simple task, and demands a careful, often time-consuming procedure, especially with over 70 thousand cries from around 3600 hours of recording. The cry editing and selection process we employed was indeed extensive but also absolutely necessary. To test our predictions regarding cry development and cause, cries were collected in real-life home settings over several 48-hr sessions. Real life is noisy. This resulted in a massive amount of audio data with unavoidable background noise which needed to be removed in order to ensure that our cries were high-quality signals for robust and reliable acoustic analysis. Such a vast number of cries cannot realistically be segmented fully manually, and some form of automated segmentation is unavoidable. Importantly, we show that we combined this automated process with thorough manual preparation, piloting, and manual verification of the automatic segmentation. This careful and systematic process took several months.

With the extensive details we have now provided alongside access to all coded data (e.g., acoustic analyses) and scripts, we trust that the entire procedure is now transparent and reproducible.

Moreover, we now provide the complete set of recordings in the form of a cry database (“*EnesBabyCriesI*”). This database contains two sets of recordings (*EnesBabyCriesIA* and *EnesBabyCriesIB*). *EnesBabyCriesIA* contains the sequences of recordings from step 3 (i.e., after removal of noisy parts, see Figure 4). *EnesBabyCriesIB* contains the cries from step 6 (after segmentation and selection of the cries; these cries are the ones used in the present

research). This databank is of course anonymized. No clues to the identity of the baby have been left in the recordings. The metadata accompanying the cries are: 1) the age of the recorded baby, 2) the baby's sex, 3) the cause of the cry as stated by the parent, 4) the action that ended the cry. We obtained parental permission to make these recordings and information public. Given the very small number of baby cry databases available, we believe this addition is a real asset to our study.

R1_20 What does a minimum of 20% voiced mean? This seems to be very low harmonicity. Baby cries typically have unvoiced regions, both subharmonic and shifts. Were unvoiced elements relevant in the "information in the cry" sought by the authors? Accordingly, exactly what was done here and why needs to be explained.

- The criterion "median pitch > 150 Hz" is obscure. Baby sounds have a median pitch of about 400 Hz. What has been evaluated here? Grunts with a lower frequency were excluded. In my opinion, the criterion must have led to analysis errors, despite the claim of "examining the ratio of false syllables". In my view, this is a serious methodological problem.

The minimum voicing threshold of 20% and f_0 of >150 Hz were employed to filter out non-cry vocalizations and any remaining remnants of background noises from the pool of automatically segmented sounds. In our sample, most cries were largely voiced (median voicing = 70% of non-silent frames), and f_0 was, as noted by the referee, typically much higher than 150 Hz (median = 430 Hz). However, these thresholds are appropriate: based on our examination of a subsample of the extracted cries, f_0 does indeed drop to 200 Hz and even lower in some infant cries, and cries do include a lot of wheezing and other unvoiced sections (not to mention that pitch trackers can simply fail to detect voicing in very quiet or very noisy sections). As noted above, this process was piloted and verified, showing only 2% of false positives.

R1_21 Does it really make sense to evaluate "jitter" (and "shimmer") when considering the recording conditions? How reliable are the values?

We agree that jitter and shimmer are highly dependent on accurate pitch tracking and correct detection of individual glottal cycles in noisy vocalizations such as baby cries. We included these two measures of voice quality for comparative purposes and, importantly, we made sure to verify the quality of pitch tracking by checking it manually in a subsample of recordings, as also explained in our response to R2_3.

R1_22 Acoustic analysis

- Again, how are the cry syllables defined, each of which is said to have been analysed with 10 key acoustic variables? What is the difference between a cry syllable and a cry (Table S2)?

We hope that the explanations provided above (and incorporated into the manuscript) have clarified the situation.

R1_23 The selection of the acoustic variables is not explained. The comment "known to be highly informative" is insufficient. The selection and significance of the acoustic measurements must be put in the context of the study's issue of concern and be supplemented with theoretical considerations. Hardly anything is said about this. The coherence of the content of the individual parts of the manuscript suffers as a consequence.

We were interested in how cries differ between individuals, how babies' voices change with age, and how cries emitted in different contexts differ acoustically. We have now added theoretical justifications for the acoustic parameters we analyzed, explaining why these are indeed the most likely candidates to encode for static and dynamic vocalizer traits and states.

The new text in the Methods now reads (lines 683-713):

“The choice of these variables was based on a large body of research in animal communication and the human voice sciences, implicating these acoustic variables as markers to speaker identity, speaker physical traits, and/or motivation and emotion (Pisanski et al., 2022; Charlton et al. 2022; Charlton et al., 2020; Kreiman & Sidtis, 2011; Pisanski & Bryant, 2019). A key acoustic parameter in human nonverbal vocal communication is fundamental frequency, perceived as voice pitch, which is highly individualistic and stable between-individuals (e.g., Fouquet et al., 2016), and yet also dynamic within-individuals and thus critical in the communication of affect and emotion (Arias et al. 2021). We hence included the following acoustic parameters: *Median Pitch* (median fundamental frequency f_0 , given in hertz) as a measure of central tendency that is more robust to noise in pitch tracking than is the mean pitch, and *Pitch IQR* (interquartile range of f_0 , in Hz), which was used instead of standard deviation or range as such measures are less sensitive to noise (i.e., incorrectly measured pitch in some voiced frames). The overall proportion of frames that are voiced was also included as it can distinguish between mostly tonal whine-like cries and wheezy or breathy vocalizations, but also because measures of voice quality were calculated specifically for voiced frames (*Voicing*, scaling from 0 to 1). Voice quality or “timbre”, understood as any acoustic characteristics that distinguishes between two voices at the same intensity and pitch, was captured by several acoustic variables including *Spectral centroid* (median spectral centre of gravity of voiced segments, in Hz), which indicates how much energy is present in high versus low frequencies and distinguishes between bright or shrill and relatively “dark” voices. Finally, several acoustic parameters were measured to describe tonal versus noisy vocalisations: *Entropy* (median Weiner entropy, scaling from 0 to 1), *Harmonics-to-noise ratio* (measure of harmonicity, in decibels), *Jitter* (short-term disturbances in f_0 , given as a percentage), *Shimmer* (short-term disturbances in the amplitude of the sound signal, given as a percentage), and *Roughness* (median proportion of modulation spectrum of voiced segments within the roughness range of amplitude modulation 30-150 Hz, given as a percentage). Working with short cries, temporal structure cannot be captured, apart from one obvious descriptive – cry *Duration* (in seconds). Amplitude (loudness) of cries could not be computed because the recording distance (microphone to baby) was not perfectly standardized. Acoustic measurements were performed in *soundgen 2.0.0* (Anikin, 2019), except for *jitter* and *shimmer* that were measured in Praat (Boersma & Weenink, 2021).”

We have also added additional details to beginnings of the Results (lines 217-225):

“We first tested whether individual cries, regardless of cry context, carry acoustic information about the vocalizing infant’s sex and age based on ten acoustic predictors known to be important in human and animal nonverbal communication: average fundamental frequency (median pitch) and its interquartile range (pitch IQR), entropy, roughness, spectral centroid, harmonicity, jitter, shimmer, duration, and voicing (see Methods for a detailed description of the measured acoustic variables and their justification; see also Charlton et al., 2020; Kreiman & Sidtis,

2011; Pisanski & Bryant, 2019). Together, these acoustic parameters capture the most biologically and perceptually relevant cry characteristics including voice pitch and its variability, voice quality, and duration.”

RESULTS

R1_24 To better assess the results, the outcome data of the acoustic measurements would be very helpful. However, I suspect that these are not available.

All acoustic measurements, both per frame and summarized per cry, are available in the supplements on OSF (<https://osf.io/ru7na/> file “data.zip”).

R1_25 Depending on age, acoustic entropy and harmonicity were found to be most important. What is meant by acoustic entropy? Could you provide typical examples?

“Entropy” is Weiner entropy of the spectrum of a frame (the ratio of geometric mean to algebraic mean of the spectrum), which is close to 0 for pure tones or tonal sounds and close to 1 for white noise (see the documentation of *soundgen* function “analyze”). Thus, sounds with high entropy are relatively noisy, and sounds with low entropy are relatively tonal.

R1_26 Which parameters determined identity in the sample?

As mentioned above, we now describe in more detail the acoustic parameters that best predict the identity of a baby (R1_6).

R1_27 In the introduction, page 3, lines 74-76, it is stated: “For example, human infant cries are often characterized by nonlinear phenomena that arise from aperiodic vibration of the vocal folds and that contribute to the perceived roughness of vocalizations (Fitch et al., 2002).” The paper by Mende et al. (1990), who first described those phenomena, requires citation here: “Bifurcations and chaos in newborn infant cries.” (Physics Letters A, Volume 145, Issues 8–9, 30 April 1990, Pages 418-424).

This relevant paper by Mende et al. (1990) is now cited.

R1_28 Overall, to demonstrate that this is a solidly designed, theoretically well-grounded study, thorough revision is strongly warranted.

We hope to have met the expectations of the referee in our thorough revision of the manuscript following the referee’s suggestions, and thank the referee for their thoughtful evaluation and support.

Reviewer #2 (Remarks to the Author):

R2_1 I'm pleased to have read this submission and have no major concerns regarding the validity of its conclusions, being primarily that 0-4 mo old infant cries signal their identities, and even their distinctive ways of aging, but not their proximal causes (e.g., hunger, isolation, physical discomfort). These findings depend on algorithm-based explorations of a large dataset of cry 'syllables,' relying on an acoustic synthesis program for acoustic parameter extraction and on sophisticated and innovative Random Forest-based analyses of these parameters. The cause observations were also corroborated perceptually, finding that virtually no gender/experience category of listener reliably detected these causes. The author group had previously found that similar groups of listeners were able to identify babies (though I recommend that it's findings be highlighted more so in this report's discussion, as so closely complementary to the current findings).

We are very grateful to the referee for their analysis of the manuscript and their comments. We have followed the referee's advice and now highlight the results we obtained in previous studies (lines 463-470):

“In two previous studies, our team showed that this individual cry signature and the possibility of its identification by caregivers are already present at birth (Bouchet et al., 2020) and at 2.5 months of age (Gustafsson et al., 2013). In particular, we previously showed that the ability to identify a baby by their cry is independent of the sex of the listener, as well as their parental status. Both women and men are able to identify a crying baby after a short exposure to its cries, but people with previous experience with babies perform better. Here, we show that this static information carried by the baby's cry predictably drifts over time...”

R2_2 In lauding these conclusions above, however, I also allude to some concerns with the reporting, and underlying procedures. These concerns arise from my professional perspective as an acoustician closely concerned with human vocal development (and, to further qualify my remarks, my focus has been on non-cry speech-like vocal development in the first year of life). I respect that the author group and the literature cited here is primarily ethological, but to reach and align with human speech development research I strongly recommend that they reconsider use of the term 'syllable' in reference to their cry units. Similarly respectful of the authors' intended contributions, a limitation towards not just delivering clear information to developmentalists in terms that may link to underlying behaviors and physiologies, it must be acknowledged that the reliance on an automatic algorithm (with no apparent validation against any ground truth measures or 'gold standard' human coding), does not permit a clear definition of the units measured here. That being said, 'continuous non-background sound' probably associates with exhalatory breath groups (though possibly including rhythmic inhalatory/exhalatory cycles, but probably excluding many extended glottal holds, i.e., breaking the surrounding phonation into distinct units). Regardless of whether these speculations are grounded (though I'm confident that anyone listening to the units could gain a good impression), these are not syllables. The onset of marginal and canonical syllables, specifically involving consonant-vowel or vowel-consonant transitions, all much shorter than the range of durations reported here, is arguably the single most important milestone in human vocal development. In the present case, because I found no discussion of any theoretical or ontological basis for the cry unitizing, I sense that this is actually a minor issue of nomenclature. Suggested short substitutes for breath group might be phrase, utterance, vocalization (or even simply 'cry' and 'cries,' with no further ascription). Returning to the broad application of algorithms, first for parameter extraction and secondly for data analyses,

I need to acknowledge that I have no direct expertise with the analysis approaches; The outcomes are adequately presented and reasonably interpretable in statistical and graphic forms.

We understand the referee's point. The term "syllable" is commonly used in bioacoustics to designate a sound framed by two silences (for example, in a bird song), but it is true that in human vocal production this term has a more restrictive meaning and can be misinterpreted. As suggested by the referee, we have therefore changed it simply to "cry" and "cries" throughout the manuscript.

R2_3 Regarding the acoustic analyses, however, I felt all could have been more carefully couched in terms of the limitations imposed by algorithmic approaches and their own control parameters. It's a bit regrettable, that though they did use the well-documented and authoritative phonetic analysis program Praat for some parameter extractions, the authors relied instead on a synthesis package for most. Yet with any program, constraints on an f_0 extraction algorithm (PDA), will have a significant impact on the representativeness and distribution characteristics of the dataset. Of greatest relevance here, the preponderance of non-modal vibratory regimes found in infant phonation literally begs questions of definition. Considering that the range of possible glottal periodicities is not just on the order of multiple kHz but also down to 10's of Hz, f_0 defined as such requires supervision of PDAs. That was clearly not the case here, perceived pitch is a completely different level of interpretation, yet considering chaotic, subharmonic, and creaky voices the authors recognize as characteristic of crying (especially early crying), f_0 data obtained in the manner here can't be interpreted as reflecting actual behavioral central tendencies or ranges. In fact, without proper parameter adjustments, the pressed nature of low frequency pulses falls below floor values and is then misread at quite high frequencies due to the high attack rate of these pulses, yielding measurement tendencies that move in exactly the wrong directions. This may even have something to do with the counterintuitive finding that f_0 increased for most of these infants with age is counterintuitive, especially in light of relatively impressionist remarks in the manuscript that older cries were less 'shrill.' My simplest request is that the authors include statements regarding limits of the approach that affect interpretation. Yet ideally, the identification of some ground truth criterion may help future efforts, calibrating against effects due to algorithmic defaults, and stepping more closely to clinical desirable understanding of the underlying physiology, cf. (Hirschberg et al., 2008).

To validate acoustic measurements and choose the most suitable approach, we analysed 100 selected individual cries with Praat and *soundgen* and then used a Praat script with manual pitch contour correction to verify automatic f_0 measurements for the same cries. Manual Praat mean f_0 and automatic Praat mean f_0 correlated with Pearson's $r = 0.55$, while manual Praat mean f_0 and automatic *soundgen* median f_0 correlated with $r = 0.75$. Therefore, we used *soundgen* pitch tracking as it was indeed more reliable. Likewise, jitter and shimmer were verified in Praat, and manual and automatic jitter measurements were found to correlate with $r = 0.79$, and shimmer with $r = 0.95$. *Soundgen* does not measure jitter and shimmer, so they could only be extracted with Praat, as already described in our Methods. These moderately high correlations indicate that automatic pitch tracking in *soundgen*, as well as jitter and shimmer measurements in Praat, are reasonably reliable, but of course not perfect considering that many cries were highly nonlinear and/or noisy. Just as with syllable segmentation, the sheer number of analyzed recordings makes some form of automatic acoustic analysis inevitable as 70,000 vocalizations cannot be checked fully manually.

R2_4 Finally, it might have been helpful to consider more carefully what speech-like utterances were captured in this dataset or how systematically they were truly eliminated—there are certainly many more such sounds than ‘grunts,’ which some take to refer to merely mechanical or ‘vegetative’ origins; there’s much related literature, too much to include, except I wish to point out just one which is so similarly in an animal ethology tradition that I think it might be considered here—I am not an author or member of this group nor involved with the prior citation: (Scheiner et al., 2002).

Hirschberg, J., Szende, T., Koltai, P., & Illenyi, A. (2008). *Pediatric Airway: Cry, Stridor and Cough*. Plural Publishing.

Scheiner, E., Hammerschmidt, K., Jurgens, U., & Zwirner, P. (2002). Acoustic analyses of developmental changes and emotional expression in the preverbal vocalizations of infants. *Journal of Voice*, 16(4), 509. <http://www.sciencedirect.com/science/article/B7585-487TGP3-27/2/3f7a034f24d6c6feded2c86cda9e96c7>

We now cite these two relevant references (lines 526 and 452). As described above (see R1_13), and also in our revised methods and new Figure 4, we made every effort to exclude non-cry sounds based on pre-defined acoustic criteria.

Reviewer #3 (Remarks to the Author):

R3_1 The research presented in the manuscript titled “What’s in a cry? Stable and dynamic information in human baby cries” aims to identify if a set of acoustic features measured in infant cry bouts can be used to accurately classify babies’ identity, age or sex, as well as the cause of the cry.

To investigate this, the authors combined modeling with psychoacoustic experiments in two samples of adults, measuring the acoustic characteristics of cries in a longitudinal sample, with babies being recorded on 48 hours measurements from 2 weeks to 4 months. The cause of the cry was also recorded.

Findings show that the models could classify above chance the identity and age of the babies starting from their cry acoustic features, but they were not able to do so for babies’ sex and for the cry cause.

Similarly, two samples of adults, both implicitly and explicitly trained could not classify the cry causes above chance. It is also important to notice that the ability to classify cries was significantly moderated by adults’ own experience with infants.

Two more interesting findings showed that (i) cries features are significantly different between individuals while stable within one individual across time and that (ii) there were some universal changes to cry characteristics with age (i.e. cry becomes more tonal).

The approach used here is innovative and the research question represents an important advance in understanding and has important practical applications. This research project helps to objectively test an impactful bit of common knowledge which, if understood better, could improve the advice given to parents as well as parents’ own experience and well-being in the first months of their baby’s life.

The statistical analysis is appropriate and the level of detail provided in the methods will allow other researchers to reproduce the work.

I recommend the manuscript for publication pending a few minor revisions (see below).

We thank the referee for their appreciation and positive evaluation of our work. We have taken into account all of the referee's comments, to which we respond in line below.

R3_2 Across the whole manuscript the authors claim that their findings show that it is not possible to classify a cry's cause based on its acoustic features alone. However, they themselves highlight that individuals' cries vary significantly and that experience with infants in general as well as with a particular infant is important in determining the cause of a cry. Moreover, the cry itself changes with age and it becomes, as written in the manuscript, more dependent on context while the baby grows. I am aware that models including the individual and the age as factors, as well as their interaction, were also run with no real improvement in the classification accuracy. However, not all the individual babies gave samples for all timepoints and this might have undermined the possibility to classify stable features in individual babies across time. Moreover, the latest time point, which could have been the one with the most structured cries, had fewer babies and this could have undermined the classification power and the most reliable – because of the cry stronger relation to context – timepoint.

We agree with the referee. It is possible that babies' cries become more contextually informative as the baby gets older. Moreover, for various reasons (the first being that babies cried less frequently as they reached 4 months of age than at younger ages), a few babies could not be recorded at all ages. In the discussion paragraph that considers this question, we have added the following text (lines 455-460):

“In our study, we did not record as many crying babies at 3.5 months as at younger ages, as babies typically cry less frequently at older ages (Barr et al., 2000), and thus a few babies were not recorded at all ages. This may have weakened the classification power of the classifiers, and it cannot thus be excluded that at 3.5 months – and a fortiori at higher ages – babies' cries may become more informative about their causes.”

R3_3 In the psychoacoustic experiments, adult participants were given some experience with a particular baby before being tested on the same baby. However, it might be necessary to have more experience with one particular baby or, alternatively, the failed classification might again be linked to age. Adult participants listened to cries of babies at 1.5 months. But again, cry might take longer to become contextualised. Therefore, results could change when considering older babies only or by testing their own parents.

We agree with the referee that several factors, such as the age of the baby or the length of the listeners' training with the babies' cries, may have influenced the listeners' abilities to identify the causes of a cry. We now discuss this point in the discussion (lines 516-521):

“It should also be noted that, although our psychoacoustic experiments did not reveal a significant ability of adult listeners to identify the cause of a cry, factors such as the age of the baby (the experiments were conducted with cries from babies aged 1.5 months only) or the

duration of the listeners' training could impact this ability. It is therefore possible that highly trained listeners, with cries from older babies, could perform better.”

R3_4 Lastly, no dynamic acoustic features (i.e. envelope, number and length of pauses, etc) were included in the analysis and these might carry important information about the type of cry or the context. The findings from the paper are solid and the authors do address some of these considerations in the limitation section. However, the language used overall in the manuscript makes the reader think that the results about the cry causes are more definite and that in general the cause of a cry cannot be determined from the cry itself ever. I would appreciate this to be smoothed out by leaving more room for doubt or for different factors.

We agree with the referee that our study - like any other study - has its limitations and that it is not impossible that we missed some information about the cause of cries. As noted by the referee, we have written a section dedicated to the limitations of our study in the discussion. We have strengthened it in this revised version. In particular, we make it clear that information may be present in longer cry sequences, whereas we only considered short cries. Concerning the ability of adult listeners to decode possible information in cries, we now also insist that the age of the baby and the level of training of the listeners are factors whose impacts remain to be explored. The limitations section now reads as follows, lines 522-556:

“The present study, while the first of its kind, has its limitations. As one cannot simply ask a preverbal baby why he or she is crying, we labeled the ostensible cause of each cry based on parental responses and behaviors that stopped the cry, and did not include rare and more extreme cry causes such as pain, wherein pain cries can be characterized by severe deviations from typical cry acoustic profiles (Bellieni et al., 2004; Koutseff et al., 2018; see also Hirschberg et al., 2009 for an analysis of pathological sounds emitted by babies). By using a parental questionnaire, we obviously took the risk that parents might have a biased assessment of their babies' cries. However, the fact that there was a 75% correspondence between the parents' assessments and what actually stopped the cry suggests that this assessment was fairly valid. We also took the precaution of considering the action that stopped the cry rather than the parental assessment if the two did not match, which did not change our pattern of results.

In order to obtain high-quality cries for acoustic analysis and perception experiments, our results are based on brief individual cries extracted from longer cry sequences. While this ensured that our cry stimuli were devoid of noise and allowed for robust acoustic measurements, it is possible that focusing on long cry sequences may have produced different results. For example, we cannot exclude the possibilities that the number and duration of pauses in cry sequences lasting several seconds or minutes may be informative features, or that information about cry cause may be encoded in the temporal, prosodic, or melodic structure of cry bouts, rather than in the spectrotemporal features of individual cries.

While our perception experiments corroborate earlier work showing that discriminating the cries of a given infant improves following brief familiarization (Gladding, 1979), the ability to recognize an infant's cry among others depends mainly on the time spent with the child (Bouchet et al., 2020). Cry recognition is thus likely to improve substantially with more extensive exposure, and possibly also with

exposure to longer cry bouts. We also highlight the fact that identification of cry context is typically multi-modal for parents, depending on visual, auditory, and olfactory cues as well as contextual cues (e.g., time of day or since feeding) that may together help parents to identify the cause of crying and resolve it in real-life contexts. Finally, as noted, the use of short cries may limit information about dynamic cry structure and temporal parameters, cues that parents may use to decide on caregiving interventions (Zeifman, 2004). It would thus be worthwhile to conduct perception experiments using longer cry sequences to test the full extent to which human listeners can effectively decode contextual information from babies' cries. Moreover, by including cries from babies of various ages in listening experiments, and varying listeners' time of exposure to those cries, researchers can test whether baby age and training intensity predict how well human listeners can decode cry cause from babies' cries."

R3_5 Discussion, line 379. The authors "predict" what an abrupt deviation from the usual acoustic feature might signal. I am not clear why a prediction is being made here in the discussion.

We have changed this sentence to explain our idea more clearly (lines 476-484):

"The presence of a unique, predictably drifting vocal signature in a baby's cries may therefore provide caregivers with a stable reference point. Any deviation from this predictable trajectory of one or more of the acoustic features defining this signature could signal the onset of a possible problem deserving special attention. It can be hypothesized that the individual signature present in the baby cry has been selected in our ancestors for this purpose. Although they lived in small groups, where the probability of confusing one baby with another was probably small, they were able to assess the baby's condition by knowing the stable characteristics of their cries."

R3_6 In the parent questionnaire on cry cause, parents were asked both their opinion on the cry cause and which action made the cry stop. In 75% of cases the two overlapped. Where they did not agree, the action that made the cry stop was considered the cry cause for the purpose of the experiment. I am wondering what would happen to the model classification accuracy of cry cause if the parental opinion was used instead. I do not really expect it to change much, but it would be a nice control to have.

As suggested by the referee, we reran the classification calculations using the parents' opinion rather than the action that stopped the cry. The results are below and, indeed, there is no substantial change. We have therefore added a sentence mentioning this in the manuscript (lines 322-325):

"Using the cause of crying indicated by the parents in the questionnaire, instead of the action that stopped crying, produced nearly identical results with a classification accuracy of 35%, 95% CI [33, 36]. This is not meaningfully different from 36% [33, 38] based on the cause that stopped crying."

R3_7 The training and testing session in the psychoacoustic experiments featured cries from the same baby. Was it the same baby for all participants?

To limit the risk of pseudo-replication, we used the cries of 7 different babies. This information is already reported in the methods and in the legend of Figure 3, but given its importance, we now also specify it in the main text (line 366): “to limit pseudo-replication, we used the cries of 7 different babies; each participant was assigned one of these 7 babies.”

R3_8 Which version of the analysis software (i.e. Praat, R) was used? It is important information for other researchers who would like to reproduce the work.

We used Praat version 6.1.16. This information is now reported in the methods (line 623). Versions of key R packages are likewise mentioned in the text.

References

All references cited here are listed in the main text reference list.

16th Jun 23

Dear Professor Mathevon,

Thank you for your patience during the peer-review process. Your manuscript titled "What's in a cry? Stable and dynamic information in human baby cries" has now been seen by the same 3 reviewers as before, and I include their comments at the end of this message. Two of the referees find your paper ready for acceptance, but one reviewer mentions some remaining hesitations. Before we make a final decision on publishing your papers, we would like to consider your responses to reviewer and editorial concerns and assess a revised manuscript.

Editorially, in addition to responding to Reviewer #2's concerns about the "ground-truth", we ask that you revise the manuscript addressing the following issues:

Please ensure that the Abstract and Introduction clearly state how the "cause" was determined (i.e. by the action that stopped crying, which in 75% of cases aligned with parents' inferred cause).

Please move the Methods section after the Introduction and before the Results. Please use the Reporting Summary as a guideline to include all necessary information in the Methods section (everything requested in the RS should also be in the Methods).

In the Results, please ensure that statistics are fully reported in the text, rather than only in Figures. Every single statement should be accompanied by appropriate statistics. For example, the sentence: "Multivariate Bayesian mixed models did not reveal any consistent acoustic differences between the cries of infant boys and girls, either overall or for any specific age group (Fig. 1a), corroborating previous work on 3-month-old babies (Reby et al., 2016)." needs statistics reported in the text.

Please avoid implicit novelty claims, i.e. stating that something has never been shown. For example, the sentence: "In particular, and to our knowledge, there is no dataset available containing cries from a cohort of babies recorded systematically and longitudinally at multiple ages during the first months of life and whose causes of crying have been labelled in a bottom-up, systematic manner." would better be rephrased for example as: "In contrast, here we present a database containing cries from a cohort of babies recorded systematically and longitudinally at multiple ages during the first months of life and whose causes of crying have been labelled in a bottom-up, systematic manner."

Further processing of your manuscript will be greatly facilitated if you ensure that it fully complies with our guidelines, as conveyed by this checklist: Communications Psychology formatting checklist (see also below). The checklist refers you to more guidance on statistics reporting and interpretation on our webpage.

Please use the following link to submit your revised manuscript, point-by-point response to the referees' comments (which should be in a separate document to any cover letter) and the completed checklist:

[link redacted]

We hope to receive your revised paper within 2-3 weeks; please let us know if you aren't able to submit it within this time so that we can discuss how best to proceed. If we don't hear from you, and the revision process takes significantly longer, we may close your file. In this event, we will still be happy to reconsider your paper at a later date, provided it still presents a significant contribution to the literature at that stage.

Please do not hesitate to contact me if you have any questions or would like to discuss these revisions further. We look forward to seeing the revised manuscript and thank you for the opportunity to review your work.

Best regards,

Jonna K. Vuoskoski

Jonna K. Vuoskoski, PhD
Editorial Board Member
Communications Psychology
orcid.org/0000-0003-0049-4373

EDITORIAL POLICIES AND FORMATTING

Editorial Policy: [Policy requirements](https://www.nature.com/documents/nr-editorial-policy-checklist.pdf) (Download the link to your computer as a PDF.)

Furthermore, please align your manuscript with our format requirements, which are summarized on the following checklist:

[Communications Psychology formatting checklist](https://www.nature.com/documents/commspsychol-style-formatting-checklist-article-rr.pdf)

and also in our style and formatting guide [Communications Psychology formatting guide](https://www.nature.com/documents/commspsychol-style-formatting-guide-accept.pdf) .

* **TRANSPARENT PEER REVIEW:** Communications Psychology uses a transparent peer review system. This means that we publish the editorial decision letters including Reviewers' comments to the

authors and the author rebuttal letters online as a supplementary peer review file. However, on author request, confidential information and data can be removed from the published reviewer reports and rebuttal letters prior to publication. If your manuscript has been previously reviewed at another journal, those Reviewers' comments would not form part of the published peer review file.

* **CODE AVAILABILITY:** All Communications Psychology manuscripts must include a section titled "Code Availability" at the end of the methods section. In the event of publication, we require that the custom analysis code supporting your conclusions is made available in a publicly accessible repository; at publication, we ask you to choose a repository that provides a DOI for the code; the link to the repository and the DOI will need to be included in the Code Availability statement. Publication as Supplementary Information will not suffice. We ask you to prepare code at this stage, to avoid delays later on in the process.

* **DATA AVAILABILITY:**

All Communications Psychology manuscripts must include a section titled "Data Availability" at the end of the Methods section or main text (if no Methods). More information on this policy, is available at <http://www.nature.com/authors/policies/data/data-availability-statements-data-citations.pdf>.

At a minimum the Data availability statement must explain how the data can be obtained and whether there are any restrictions on data sharing. Communications Psychology strongly endorses open sharing of data. If you do make your data openly available, please include in the statement:

We recommend submitting the data to discipline-specific, community-recognized repositories, where possible and a list of recommended repositories is provided at <http://www.nature.com/sdata/policies/repositories>.

If a community resource is unavailable, data can be submitted to generalist repositories such as <https://figshare.com/> or <http://datadryad.org/> Dryad Digital Repository. Please provide a unique identifier for the data (for example a DOI or a permanent URL) in the data availability statement, if possible. If the repository does not provide identifiers, we encourage authors to supply the search terms that will return the data. For data that have been obtained from publicly available sources, please provide a URL and the specific data product name in the data availability statement. Data with a DOI should be further cited in the methods reference section.

REVIEWERS' COMMENTS:

Reviewer #1 (Remarks to the Author):

I have no further comments as I am satisfied with the authors' response. Well-done work.

I support publication of the revised version.

Kathleen Wermke

Reviewer #2 (Remarks to the Author):

I appreciate the authors' adoption of the far less problematic term 'cries' as opposed to 'syllables,' and I'm confident that broad readership will as well.

I also appreciate that the authors were responsive in their rebuttal letter to my concerns regarding reliance on specific algorithms primarily from a not-widely-used synthesis program, especially the fact that they'd conducted some preliminary explorations of fo reliability. That being said, reproducibility of fo measures as assessed by correlations disregards the common occurrence of tracking may follow similar contours but at octave differences. And my primary concern remains that arbitrary algorithmic parameter settings, especially the declared fo floor value of 150, will have constrained possible variation and introduced some presence of artifacts in the data. But regarding the 150 Hz floor; it might be argued on more 'gold standard' prior datasets that crying fo only rarely dips into the lower Hz ranges that vocal development researchers find to be quite common in infants' non-cry vocalizations (such as 'growls' and 'grunts')?

I do not deprecate the power that they obtained by eschewing manual methods (and agree that they would have been completely unworkable for such a dataset), yet this remains in effect for me the same kind of problem identified by R1: the fact that this specific implementation of a set of algorithms can't be claimed to represent ground truth (no 'gold standard'), at least not regarding infant phonatory development.

Even though the authors' purpose was not to track development per se, I request more explicit disclaimers in the manuscript itself; perhaps, if they agree, stating in effect that 'manual corroboration of the specific acoustic results obtained here would be necessary in order to be confident that they align with underlying phonatory behaviors and physiologies'. A secondary point I'd made originally regarded the fact that non-cry vocalizations were excluded; this is another reason that readers may best be cautioned against developmental interpretations of the findings reported here. But as the authors acknowledge, the simple limitation to 3.5 months' range should already limit a developmental reading of this work.

I offer below a few editorial remarks along with running observations from the text regarding examples of the main issue I consider to remain outstanding.

But I first make one small suggestion here regarding the accuracy matrices in Figures 2 and 3 first: Could the graphic-number mapping effectively be kept the same across these? Anchoring these scales to the same endpoints would reinforce the authors' point regarding the much smaller context accuracies.

p. 7, line 167: The claim for baby-specific fos needs citations, especially since it appears to have been affirmed in the current project; some discussion of any such alignment with prior results would also help to ameliorate concerns regarding the authors automatic fo results.

p. 8, line 202: I invite the authors to reword this sentence to avoid implying that obtaining perceptual context judgments 'verified the robustness of our acoustic analyses'; it might better be said that this verified the robustness of 'results based on our acoustic analysis' (just not the analyses themselves).

Discussion

In the first paragraph, the authors modestly imply that perhaps context effects would be found in an even larger dataset, though the size of the dataset used here was an attribute and certainly impresses me as adequately powered, so perhaps other disclaimers may be more important here; being restricted to the first four months of life, with just 'static' markers as it were, but also again with the current results limited by the constraints associated with a certain set of unsupervised algorithms. Along such lines, the authors do in fact proceed to acknowledge that other acoustics-based approaches, e.g. dynamic contour variation, may yield context effects. Yet these will also develop with age and begin to co-mingle with the development of phonological structures, so this is not really a study limitation so much as a difficult the nature of developmental communication research.

Having said that physiological claims aren't well-supported here, I don't dismiss the maturational conjectures offered in lines 444-446, but I see here again an opportunity to emphasize that more intensive acoustics-based research would be needed to corroborate such.

Reviewer #3 (Remarks to the Author):

The authors addressed all my previous concerns, I am now happy to recommend the paper for publication.

Response to Referee 2

R2_1 I appreciate the authors' adoption of the far less problematic term 'cries' as opposed to 'syllables,' and I'm confident that broad readership will as well.

Thank you!

R2_2 I also appreciate that the authors were responsive in their rebuttal letter to my concerns regarding reliance on specific algorithms primarily from a not-widely-used synthesis program, especially the fact that they'd conducted some preliminary explorations of fo reliability. That being said, reproducibility of fo measures as assessed by correlations disregards the common occurrence of tracking may follow similar contours but at octave differences. And my primary concern remains that arbitrary algorithmic parameter settings, especially the declared fo floor value of 150, will have constrained possible variation and introduced some presence of artifacts in the data. But regarding the 150 Hz floor; it might be argued on more 'gold standard' prior datasets that crying fo only rarely dips into the lower Hz ranges that vocal development researchers find to be quite common in infants' non-cry vocalizations (such as 'growls' and 'grunts')?

Regarding the reliability of fo tracking, we compared mean fo derived from soundgen with the *corrected* mean fo derived from Praat for each vocal stimulus. That is, a trained researcher first checked the Praat fo tracker to ensure that the estimates were correct, and only then checked the correlation between the mean of this manually-verified contour and two different automatic measurements (uncorrected Praat and soundgen). Thus, the reported measure of reproducibility of fo measures is not about how well two algorithms agreed with each other and not about correlations between two contours (possibly different by an octave, as pointed out by the Reviewer), but about how well automatically measured mean fo per sound matched manual measurements.

As for the referee's second point, we cannot avoid small mistakes and artifacts in the data: it is simply not possible to perform automatic acoustic segmentation and analysis of real-life recordings, especially of such vocalizations as baby cries, without introducing some "noise". Naturally, we used all our expertise to minimize this noise, including setting what we thought were reasonable values for pitch floor and ceiling, which we further confirmed with signal detection checks (e.g., reducing the false positive rate to 2%). We respect the Reviewer's doubts, and note that both the cry recordings and the scripts for analyzing them are freely available to the public, so that any interested person can reproduce our measurements, or perhaps improve upon them.

R2_3 I do not deprecate the power that they obtained by eschewing manual methods (and agree that they would have been completely unworkable for such a dataset), yet this remains in effect for me the same kind of problem identified by R1: the fact that this specific implementation of a set of algorithms can't be claimed to represent ground truth (no 'gold standard'), at least not regarding infant phonatory development.

Please see R2_2 above.

R2_4 Even though the authors' purpose was not to track development per se, I request more explicit disclaimers in the manuscript itself; perhaps, if they agree, stating in effect that 'manual corroboration of the specific acoustic results obtained here would be necessary in order to be confident that they align with underlying phonatory behaviors and physiologies'.

A secondary point I'd made originally regarded the fact that non-cry vocalizations were excluded; this is another reason that readers may best be cautioned against developmental interpretations of the findings reported here. But as the authors acknowledge, the simple limitation to 3.5 months' range should already limit a developmental reading of this work.

We have added the suggested text to the discussion:

“Manual corroboration of the specific acoustic results obtained here would be necessary in order to be confident that they align with underlying phonatory behaviors and physiologies. In future, it will also be important to study the developmental trajectory of crying over a broader age range and using more sensitive acoustic descriptives, including pitch contours and measures of temporal dynamics within bouts of crying.”

Please also see R2_9.

R2_5 I offer below a few editorial remarks along with running observations from the text regarding examples of the main issue I consider to remain outstanding. But I first make one small suggestion here regarding the accuracy matrices in Figures 2 and 3 first: Could the graphic-number mapping effectively be kept the same across these? Anchoring these scales to the same endpoints would reinforce the authors' point regarding the much smaller context accuracies.

Thank you for this suggestion! The mapping of accuracy to color has been standardized in these two figures (Figures 3b and 4b).

R2_6 p. 7, line 167: The claim for baby-specific fos needs citations, especially since it appears to have been affirmed in the current project; some discussion of any such alignment with prior results would also help to ameliorate concerns regarding the authors' automatic fo results.

Although we gave citations to support this claim in the paragraph just above, we have now also added a citation to this statement: “Although f_0 is known to differ across babies (Reby et al., 2016)...”

R2_7 p. 8, line 202: I invite the authors to reword this sentence to avoid implying that obtaining perceptual context judgments ‘verified the robustness of our acoustic analyses’; it might better be said that this verified the robustness of ‘results based on our acoustic analysis’ (just not the analyses themselves).

Thank you, corrected!

R2_8 Discussion. In the first paragraph, the authors modestly imply that perhaps context effects would be found in an even larger dataset, though the size of the dataset used here was an attribute and certainly impresses me as adequately powered, so perhaps other disclaimers may be more important here; being restricted to the first four months of life, with just ‘static’ markers as it were, but also again with the current results limited by the constraints associated with a certain set of unsupervised algorithms. Along such lines, the authors do in fact proceed to acknowledge that other acoustics-based approaches, e.g. dynamic contour variation, may yield context effects. Yet these will also develop with age and begin to co-mingle with the

development of phonological structures, so this is not really a study limitation so much as a difficult the nature of developmental communication research.

The phrase in question is probably “[cries] do not communicate robust information about their cause, at least not within our very large sample of brief individual cries”. Here we are rather emphasizing that the cries were brief and individual, thus referring to the lack of dynamic cry data. We also meant that we failed to discover such differences, although the analyzed sample was in fact very large. We certainly agree with the reviewer that it would be more fruitful to try other approaches to studying the developmental trajectory of crying, and not merely to scale the datasets even further, as we discuss in the following paragraphs.

R2_9 Having said that physiological claims aren’t well-supported here, I don’t dismiss the maturational conjectures offered in lines 444-446, but I see here again an opportunity to emphasize that more intensive acoustics-based research would be needed to corroborate such.

We added this text at the end of the paragraph in question:

“In future, it will be important to study the developmental trajectory of crying over a broader age range and using more sensitive acoustic descriptives, including pitch contours and measures of temporal dynamics within bouts of crying.”

19th Jul 23

Dear Professor Mathevon,

Your manuscript titled "What's in a cry? Stable and dynamic information in human baby cries" has now been seen by the reviewer who previously had some remaining concerns. Their comments appear below. In light of their advice I am delighted to say that we are happy, in principle, to publish a suitably revised version in Communications Psychology under the open access CC BY license (Creative Commons Attribution v4.0 International License).

We therefore invite you to revise your paper one last time to address a list of editorial requests. At the same time we ask that you edit your manuscript to comply with our format requirements and to maximise the accessibility and therefore the impact of your work.

EDITORIAL REQUESTS:

SUBMISSION INFORMATION:

OPEN ACCESS:

Communications Psychology is a fully open access journal. Articles are made freely accessible on publication under a [CC BY license](http://creativecommons.org/licenses/by/4.0) (Creative Commons Attribution 4.0 International License). This license allows maximum dissemination and re-use of open access materials and is preferred by many research funding bodies.

For further information about article processing charges, open access funding, and advice and support from Nature Research, please visit <https://www.nature.com/commspsychol/article-processing-charges>

At acceptance, you will be provided with instructions for completing this CC BY license on behalf of all authors. This grants us the necessary permissions to publish your paper. Additionally, you will be asked to declare that all required third party permissions have been obtained, and to provide billing information in order to pay the article-processing charge (APC).

* **DATA AVAILABILITY:**

[link redacted]

Best regards,

Marike

Marike Schiffer, PhD
Chief Editor
Communications Psychology

REVIEWERS' COMMENTS:

Reviewer #2 (Remarks to the Author):

I'm quite satisfied that the authors addressed my remaining concerns, and adopted some specific suggestions.